# Do We Really Need Permutations? Impact of Model Width on Linear Mode Connectivity

**Akira Ito**[1*]**, Masanori Yamada**[2*]**, Daiki Chijiwa**[3] **& Atsutoshi Kumagai**[3]
[1]Tohoku University  [2]NTT DOCOMO, INC.  [3]NTT Computer and Data Science Laboratories
akira.ito.b1@tohoku.ac.jp,
{masanori.yamada, daiki.chijiwa, atsutoshi.kumagai}@ntt.com

## ABSTRACT

Recently, Ainsworth et al. empirically demonstrated that, given two independently trained models, applying a parameter permutation that preserves the input–output behavior allows the two models to be connected by a low-loss linear path. When such a path exists, the models are said to achieve linear mode connectivity (LMC). Prior studies, including Ainsworth et al. (2023), have reported that achieving LMC requires not only an appropriate permutation search but also sufficiently wide models (e.g., a $32 \times$ width multiplier for ResNet-20). This is broadly believed to be because increasing the model width ensures a large enough space of candidate permutations, increasing the chance of finding one that yields LMC. In this work, we empirically demonstrate that, *even without any permutations,* simply widening the models is sufficient for achieving LMC when using a suitable softmax temperature calibration. We further explain why this phenomenon arises by analyzing intermediate layer outputs. Specifically, we introduce layerwise exponentially weighted connectivity (LEWC), which states that the output of each layer of the merged model can be represented as an exponentially weighted sum of the outputs of the corresponding layers of the original models. Consequently the merged model's output matches that of an ensemble of the original models, facilitating LMC. To the best of our knowledge, this work is the first to show that widening the model not only facilitates *nonlinear* mode connectivity, as suggested in prior research, but also significantly increases the possibility of achieving *linear* mode connectivity.

## 1 INTRODUCTION

Large neural networks (NNs) are widely used across various domains (Vaswani et al., 2017; van den Oord et al., 2016; Zhao et al., 2023), and optimizing their parameters constitutes a massive non-convex optimization problem. Remarkably, stochastic gradient descent (SGD), which is widely employed for NN training, is known to find good solutions despite its simplicity. One hypothesis proposed to explain this seemingly counterintuitive phenomenon is that the landscape of the loss function may be far simpler than previously thought. Several studies (Garipov et al., 2018; Draxler et al., 2018; Freeman & Bruna, 2017) have reported that different NN solutions can be connected through simple nonlinear paths with almost no increase in loss. Recently, Entezari et al. (2022) conjectured that, after accounting for all permutation symmetries in NNs, Conjecture 1.1 may hold.

**Conjecture 1.1** (Permutation invariance, informal)**.** *Let $\boldsymbol{\theta}_a$ and $\boldsymbol{\theta}_b$ be the parameters of two models. When their model widths are sufficiently large, there exists a permutation $\pi$ such that the barrier between $\boldsymbol{\theta}_a$ and $\pi(\boldsymbol{\theta}_b)$ (as defined in Definition 2.1) becomes sufficiently small with high probability.*

Here, the barrier refers to the amount of loss increase when linearly interpolating between the weights of the two models. When the barrier between them is sufficiently small, they are said to exhibit linear mode connectivity (LMC) (Frankle et al., 2020). Conjecture 1.1 claims that many SGD solutions can be mapped into the same loss basin by applying an appropriate permutation. Indeed, several studies (Ainsworth et al., 2023; Singh & Jaggi, 2020) have experimentally validated this conjecture across a variety of datasets and models by employing effective permutation-finding techniques.

---

*Work done while at NTT Social Informatics Laboratories.

In previous studies (Entezari et al., 2022; Ainsworth et al., 2023; Singh et al., 2024; Ito et al., 2025a;b; Ferbach et al., 2024), it has been widely believed that sufficiently widening the model is required to find a permutation under which LMC holds. Intuitively, if the model is not sufficiently wide, the number of candidate permutations is limited, making it difficult to discover an appropriate permutation that brings the models into the same loss basin. For example, Ainsworth et al. (2023) empirically demonstrated that, without increasing ResNet-20's width by $32\times$ or VGG-16's by $4\times$, permutations fail to sufficiently reduce the loss barrier on the CIFAR-10 dataset. In addition, several studies have shown that widening ResNet-50 trained on the ImageNet dataset improves the test accuracy of the merged model when interpolating the weights of two trained models with permutations (Ainsworth et al., 2023). The original conjecture (Conjecture 1.1) also suggests that there may be no permutation that enables LMC unless the model is sufficiently wide.

In this paper, we empirically show that once the model is sufficiently wide, simply averaging the weights of independently trained models *without applying any permutation* achieves test accuracy comparable to that of the original models. This finding implies that even without aligning models to the same loss basin via permutation, sufficiently widening the models naturally places them within the same basin in terms of test accuracy. Prior works (Nguyen, 2019; Shevchenko & Mondelli, 2020) have demonstrated that increasing model width facilitates the existence of nonlinear low-loss paths between trained models. However, to the best of our knowledge, no study has suggested that it also facilitates connectivity through linear paths. Our results suggest that widening the model itself may play a more critical role in achieving LMC than increasing the number of possible permutations.

Understanding the principle behind LMC is important not only for theoretical reasons, such as explaining the effectiveness of SGD in deep learning, but also for practical applications like model merging. Prior work (Singh & Jaggi, 2020; Wang et al., 2020; Guerrero-Peña et al., 2023) has leveraged permutation symmetries in neural networks for model merging, federated learning, and continual learning. While weight averaging is known to work well for models fine-tuned from a shared foundation model, merging independently trained models remains challenging. By studying LMC without permutations, we suggest that a similar strategy may also be feasible in this setting.

**Contributions**    The contributions of this paper are threefold:

**1. Widening improves the performance of merged models.**    We empirically show that just increasing the width of independently trained models monotonically improves the accuracy of their merged model without permutations, eventually matching the performance of the original models. Furthermore, we show that the loss barrier can be reduced to nearly zero by calibrating the softmax with an appropriate inverse temperature, thereby achieving LMC in this setting.

**2. Revealing why increasing model width facilitates LMC.**    We introduce **layerwise exponentially weighted connectivity (LEWC)**, which states that the intermediate-layer outputs of the merged model can be expressed as an exponentially weighted average of the corresponding outputs of the two original models. Since LEWC implies that the merged model behaves like an ensemble of the two models in terms of predictive performance, it explains why LMC holds. We show that widening the model makes it more likely for LEWC to hold.

**3. The role of low-rank structure in achieving LMC.**    We further show that the low-rank structure of weight matrices plays a crucial role in achieving LEWC. Previous work on permutation-based model merging has pointed out that low-rankness of weights is essential for LMC (Ito et al., 2025a; Yunis et al., 2024). We demonstrate that this requirement also applies to LEWC by conducting experiments that vary the degree of weight decay. These results suggest that the possibility for LMC to hold depends strongly on properties of the solutions obtained by SGD.

## 2  BACKGROUND AND PRELIMINARIES

### 2.1  NOTATION

For a natural number $k \in \mathbb{N}$, we denote $[k] := \{1, 2, \ldots, k\}$. We use bold uppercase letters (e.g., $\boldsymbol{X}$) for tensors and matrices, and bold lowercase letters (e.g., $\boldsymbol{x}$) for vectors. Given a tensor $\boldsymbol{X}$, its vectorized form is written as $\mathrm{vec}(\boldsymbol{X})$, and $\|\boldsymbol{X}\|$ indicates its Frobenius ($L^2$) norm.

Throughout this paper, we consider multilayer perceptrons (MLPs) $f(\boldsymbol{x}; \boldsymbol{\theta})$ with $L$ layers, though the analysis can be extended to other neural architectures. The input is $\boldsymbol{x} \in \mathbb{R}^{d_{\text{in}}}$, and the parameter set is $\boldsymbol{\theta} \in \mathbb{R}^{d_{\text{pa}}}$, where $d_{\text{in}}$ and $d_{\text{pa}}$ denote the input and parameter dimensions, respectively. Let $\boldsymbol{z}_\ell$ be the representation at the $\ell$-th layer, defined recursively as $\boldsymbol{z}_0 = \boldsymbol{x}$ and $\boldsymbol{z}_\ell = \sigma(\boldsymbol{W}_\ell \boldsymbol{z}_{\ell-1} + \boldsymbol{b}_\ell)$ for $\ell \in [L]$, where $\sigma$ is the activation, and $\boldsymbol{W}_\ell, \boldsymbol{b}_\ell$ are the weight and bias of layer $\ell$. When parameters need to be emphasized, we denote the $\ell$-th layer's output as $f_\ell(\boldsymbol{x}; \boldsymbol{\theta})$. The full parameter vector can be expressed as $\boldsymbol{\theta} = \bigoplus_{\ell=1}^{L} (\text{vec}(\boldsymbol{W}_\ell) \oplus \boldsymbol{b}_\ell)$ where $\oplus$ denotes concatenation.

## 2.2 Linear Mode Connectivity (LMC)

Let $\boldsymbol{\theta} \in \mathbb{R}^{d_{\text{pa}}}$ be a model and $\mathcal{L}(\boldsymbol{\theta})$ its loss. For two models $\boldsymbol{\theta}_a$ and $\boldsymbol{\theta}_b$, the *loss barrier* is defined as:

**Definition 2.1** (Loss Barrier (Entezari et al., 2022))**.**

$$B(\boldsymbol{\theta}_a, \boldsymbol{\theta}_b) := \max_{\lambda \in [0,1]} \Big( \mathcal{L}(\lambda \boldsymbol{\theta}_a + (1-\lambda)\boldsymbol{\theta}_b) - \big[\lambda \mathcal{L}(\boldsymbol{\theta}_a) + (1-\lambda)\mathcal{L}(\boldsymbol{\theta}_b)\big] \Big).$$

Intuitively, $B(\boldsymbol{\theta}_a, \boldsymbol{\theta}_b)$ measures how much the loss increases when linearly interpolating between the two models. If this barrier is nearly zero, we say $\boldsymbol{\theta}_a$ and $\boldsymbol{\theta}_b$ are *linearly mode connected*.

## 2.3 Permutation Symmetry and LMC

Neural networks exhibit permutation symmetry in their parameter space. For the $\ell$-th layer of a network with parameters $\boldsymbol{\theta}$, permuting its outputs and compensating at the next layer leaves the input–output behavior unchanged. We denote this transformation by $\pi(\boldsymbol{\theta})$, where $\pi$ is a permutation.

Ainsworth et al. (2023) showed that two independently trained wide networks can achieve LMC by permutation symmetries through weight matching (WM), which finds permutations that minimize their $L^2$ distance. Intuitively, if $\boldsymbol{\theta}_a \approx \pi(\boldsymbol{\theta}_b)$ for some $\pi$, then $\boldsymbol{\theta}_a$ and $\pi(\boldsymbol{\theta}_b)$ can be seen as almost the same parameters, so their interpolation $\lambda \boldsymbol{\theta}_a + (1-\lambda)\pi(\boldsymbol{\theta}_b)$ should preserve performance, where $\lambda$ is the merging (interpolation) coefficient. This view suggests that models scattered in parameter space may lie in a common loss basin once permutations are accounted for. However, prior works (Ainsworth et al., 2023; Entezari et al., 2022; Guerrero-Peña et al., 2023; Ito et al., 2025a) indicate that WM requires very large widths (e.g., $32\times$ for ResNet-20, $4\times$ for VGG-16). In contrast, our results show that simply widening the models already improves merged performance monotonically, eventually matching the originals without permutation.

## 3 Width Expansion Facilitates High-Accuracy Model Merging

Ainsworth et al. (2023) experimentally demonstrated that, in permutation-based model merging, sufficiently wide models are required for LMC to hold. In this section, we empirically show that even without permutations, widening the models improves the performance of merged models. First, we confirm that widening increases test accuracy and reduces test loss. Next, we demonstrate that applying random permutations does not degrade performance. These results suggest that, once the models are sufficiently wide, permutations are no longer essential for achieving LMC.

### 3.1 Test Accuracy and Loss

Figure 1 shows the mean and standard deviation of test accuracies over three independent merges of two models.[1] For comparison, we also include results obtained with the permutations discovered by weight matching (WM) in Figure 16. Details of the models, hyperparameters, and permutation search methods are provided in Appendix B. As shown in the figure, widening the models enables merged models to achieve accuracy comparable to the original ones, even without permutations.

Figure 2 presents the test losses of merged models. As observed in Figure 2(a), the test loss does not decrease sufficiently to match that of the original models, although Figure 1 shows that test accuracy

---

[1] Because LMC holds only when the models are sufficiently widened, our experiments are limited to relatively simple datasets such as CIFAR-10 and MNIST. In Appendix F.1, we also provide results on a more complex dataset, CIFAR-100, confirming that widening the model similarly removes the barrier.

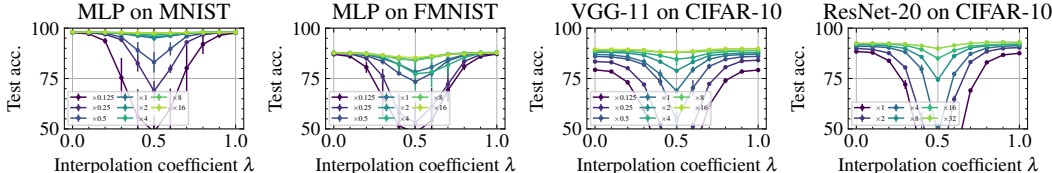

Figure 1: Test accuracies of merged models **without permutations** for different values of the interpolation coefficient $\lambda$. Even in the absence of permutations, increasing the width multiplier enables the merged models to reach accuracy comparable to the original models (corresponding to $\lambda = 0$ and 1).

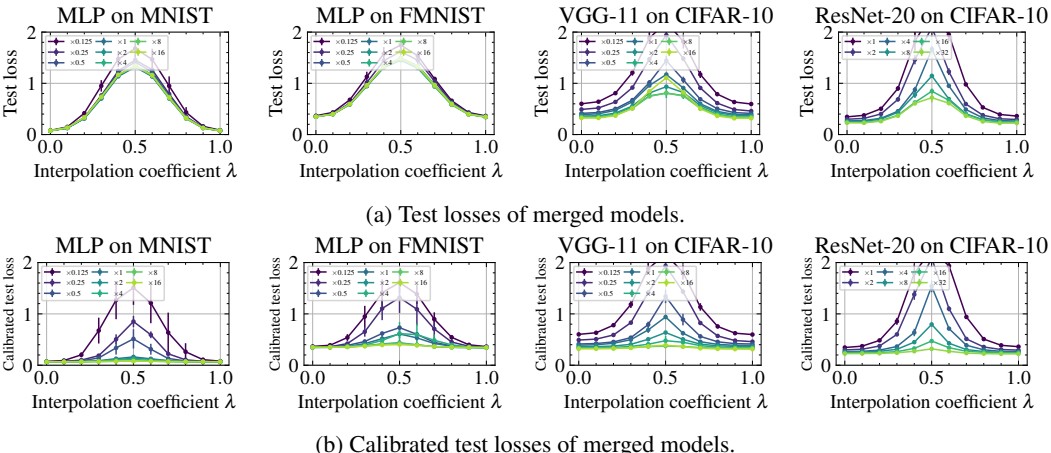

(a) Test losses of merged models.

(b) Calibrated test losses of merged models.

Figure 2: Test losses of merged models without permutations. Figure 2(a) shows the original loss values, while Figure 2(b) shows the values obtained by applying temperature scaling.

Table 1: Barrier values ($\lambda = 1/2$) with and without the permutation found by WM. Loss barriers use calibrated losses in both settings. Width multipliers are $16\times$ for MLP, $16\times$ for VGG-11, and $32\times$ for ResNet-20. For sufficiently wide models, barriers are small even without the permutation.

| Network | Dataset | Without perm | | With perm | |
|---|---|---|---|---|---|
| | | Acc. barrier (%) | Loss barrier | Acc. barrier (%) | Loss barrier |
| MLP | MNIST | $0.519 \pm 0.225$ | $0.013 \pm 0.004$ | $-0.027 \pm 0.139$ | $-0.003 \pm 0.003$ |
| | FMNIST | $2.467 \pm 0.685$ | $0.056 \pm 0.014$ | $4.925 \pm 4.379$ | $0.155 \pm 0.111$ |
| VGG-11 | CIFAR-10 | $1.308 \pm 1.594$ | $0.066 \pm 0.054$ | $7.000 \pm 3.694$ | $0.177 \pm 0.039$ |
| ResNet-20 | | $2.694 \pm 1.493$ | $0.087 \pm 0.051$ | $5.135 \pm 2.935$ | $0.173 \pm 0.099$ |

monotonically increases with width. This discrepancy arises because certain transformations of the output distribution can change the loss without affecting accuracy. A concrete example is scaling the logits by an inverse temperature, which alters the cross-entropy loss but leaves the predicted labels unchanged. Since accuracy is the primary objective in classification, it is reasonable to evaluate the loss under an optimal inverse temperature. Accordingly, we estimated the inverse temperature using 20% of the test set and computed calibrated losses on the remaining 80%. The calibrated results are shown in Figure 2(b). With this adjustment, the loss barrier approaches zero as width increases, as expected. In this sense, when we state that LMC holds with a zero loss barrier, we also include the case where the softmax is calibrated with an inverse temperature. For a more detailed view, Table 1 reports the barrier values separately for the cases with and without applying the permutation. We find that the barriers remain very small even without the permutation.

## 3.2 RANDOM PERMUTATIONS

As discussed above, sufficiently wide models achieve performance comparable to the originals even without permutations. This suggests that permutations are not essential once the models are sufficiently wide. To test this further, we applied random permutations before merging two models at $\lambda = 1/2$. As shown in Figure 18(a), the merged models still maintain high accuracy, indicating that permutations are not critical when models are sufficiently wide.

## 4 EXPANDING MODEL WIDTH ACHIEVES LAYERWISE EXPONENTIALLY WEIGHTED CONNECTIVITY

In this section, we aim to explain why widening models facilitates LMC. To this end, we introduce the concept of *layerwise exponentially weighted connectivity* (LEWC) in Section 4.1. When LEWC holds, the intermediate outputs of the merged model can be expressed as exponentially weighted combinations of the corresponding outputs from the original models. Consequently, the output of the merged model becomes equivalent to that of an ensemble of the original models, thereby achieving LMC. We then empirically examine in Section 4.2 whether LEWC actually holds at the merging ratio $\lambda = 1/2$, and show that widening the model makes LEWC more likely to be satisfied. A more fundamental explanation of why LEWC emerges will be provided in the next section.

### 4.1 LAYERWISE EXPONENTIALLY WEIGHTED CONNECTIVITY

To clarify why widening models enables LMC, we introduce the following key concept.

**Definition 4.1** (Layerwise Exponentially Weighted Connectivity). Two models with parameters $\boldsymbol{\theta}_a$ and $\boldsymbol{\theta}_b$ are said to be *layerwise exponentially weighted connected* if, for every layer $\ell \in [L]$ and any $\lambda \in [0, 1]$, we have

$$f_\ell(\boldsymbol{x}; \lambda\boldsymbol{\theta}_a + (1-\lambda)\boldsymbol{\theta}_b) = \lambda^\ell f_\ell(\boldsymbol{x}; \boldsymbol{\theta}_a) + (1-\lambda)^\ell f_\ell(\boldsymbol{x}; \boldsymbol{\theta}_b) \quad \text{almost surely.} \tag{1}$$

When Definition 4.1 holds, the intermediate output of the merged model $\boldsymbol{\theta}_c = \lambda\boldsymbol{\theta}_a + (1-\lambda)\boldsymbol{\theta}_b$ at layer $\ell$ is expressed as a weighted sum of the original models' outputs, where the coefficients decay exponentially with depth as $\lambda^\ell$ and $(1-\lambda)^\ell$. This also applies to the last layer, yielding $f_L(\boldsymbol{x}; \boldsymbol{\theta}_c) = \lambda^L f_L(\boldsymbol{x}; \boldsymbol{\theta}_a) + (1-\lambda)^L f_L(\boldsymbol{x}; \boldsymbol{\theta}_b)$. For classification tasks, since scaling the logits by a positive constant does not change the predicted labels, we divide the right-hand side by $\lambda^L + (1-\lambda)^L$ to normalize the coefficients into weights that sum to one. In this way, the merged model can be interpreted as an ensemble that uses a weighted average of the two models' logits with weights $\lambda^L/(\lambda^L + (1-\lambda)^L)$ and $(1-\lambda)^L/(\lambda^L + (1-\lambda)^L)$. Thus, in terms of accuracy, LEWC directly implies LMC (i.e., no accuracy degradation).

On the other hand, for the loss, when $\lambda$ is close to $1/2$, the exponential decay factors $\lambda^L$ and $(1-\lambda)^L$ may cause the loss to increase. However, this effect can be mitigated by applying a temperature-scaled softmax with an appropriate inverse temperature. As shown in Figure 2(b), with suitable calibration the loss barrier can also be reduced to nearly zero.

**Relation to layerwise linear feature connectivity.** Zhou et al. (2023) introduced *layerwise linear feature connectivity* (LLFC), a concept related to LEWC. LLFC explains why LMC holds for permutation-based methods and spawning. LLFC states that, for each layer, the output of the merged model can be expressed as a weighted average of the outputs of the two original models. As sufficient conditions for LLFC, Zhou et al. (2023) proposed *weak additivity for ReLU activations* and a *commutativity property*. In our setting, we show in Appendix C that the commutativity property does not hold. This is why, in this work, we introduce LEWC as a concept distinct from LLFC.

### 4.2 EMPIRICAL VERIFICATION

We next empirically verify whether LEWC holds at $\lambda = 1/2$. In this case, Equation (1) reduces to $f_\ell(\boldsymbol{x}; (\boldsymbol{\theta}_a + \boldsymbol{\theta}_b)/2) = (1/2)^\ell(f_\ell(\boldsymbol{x}; \boldsymbol{\theta}_a) + f_\ell(\boldsymbol{x}; \boldsymbol{\theta}_b))$. Accordingly, we measured the cosine similarity between $f_\ell(\boldsymbol{x}; (\boldsymbol{\theta}_a + \boldsymbol{\theta}_b)/2)$ and $(f_\ell(\boldsymbol{x}; \boldsymbol{\theta}_a) + f_\ell(\boldsymbol{x}; \boldsymbol{\theta}_b))/2$, where the factor $(1/2)^\ell$ can be omitted since cosine similarity is invariant to positive scaling. The results are shown in Figure 3.

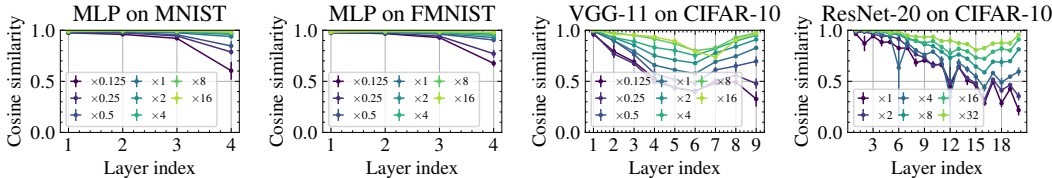

Figure 3: Average cosine similarity between $f_\ell(\boldsymbol{x}; (\boldsymbol{\theta}_a + \boldsymbol{\theta}_b)/2)$ and $(f_\ell(\boldsymbol{x}; \boldsymbol{\theta}_a) + f_\ell(\boldsymbol{x}; \boldsymbol{\theta}_b))/2$ for each layer when test data are fed into the models. For the last layer, cosine similarity is computed between the logits. The color of each plot indicates the degree of width expansion. Wider models exhibit higher cosine similarity, making it easier for LEWC to hold.

Across all models, increasing the width consistently improves cosine similarity at each layer. In particular, when the width is sufficiently large, the cosine similarity at the last layer becomes close to 1, indicating that the merged model's output is almost identical to that of the ensemble of the two original models. This explains why the merged model achieves high test accuracy.

## 5 WHY DOES LEWC EMERGE IN THE WIDE-WIDTH REGIME?

In this section, we clarify why widening the model leads to the satisfaction of LEWC. First, in Section 5.1, we introduce, as sufficient conditions for LEWC, (1) weak additivity for ReLU activations and (2) reciprocal orthogonality. When (1) and (2) are satisfied, LEWC holds. Therefore, in Sections 5.2 and 5.3, we empirically verify whether conditions (1) and (2) are satisfied, and further reveal that the reasons for their validity are due to the low-rank structure of the weights. In other words, conditions (1) and (2) do not hold when the weights are not low-rank, in which case neither LEWC nor LMC holds. Previous studies (Ito et al., 2025a; Galanti et al., 2022) have pointed out that weakening weight decay during SGD training increases the rank of the weights. Therefore, in Section 5.4, we empirically demonstrate that by indirectly increasing the rank of the weight matrices through weakening weight decay, both LEWC and LMC do not hold. This clarifies that the rank of the weight matrices strongly influences the realization of LEWC and LMC.

### 5.1 SUFFICIENT CONDITIONS FOR LEWC

In this section, we introduce, as sufficient conditions for LEWC, (1) weak additivity for ReLU activations and (2) reciprocal orthogonality. First, (1) is defined as follows.

**Definition 5.1** (Weak Additivity for ReLU Activations (Zhou et al., 2023)). Let $\tilde{\boldsymbol{z}}_\ell^{(a)}$ and $\tilde{\boldsymbol{z}}_\ell^{(b)}$ be the $\ell$-th layer pre-activations in two models, respectively. These models satisfy *weak additivity for ReLU activations* if, for every layer $\ell \in [L]$ and any $\lambda \in [0, 1]$, $\sigma(\lambda \tilde{\boldsymbol{z}}_\ell^{(a)} + (1 - \lambda)\tilde{\boldsymbol{z}}_\ell^{(b)}) = \lambda \sigma(\tilde{\boldsymbol{z}}_\ell^{(a)}) + (1 - \lambda)\sigma(\tilde{\boldsymbol{z}}_\ell^{(b)})$ almost surely, where $\sigma$ denotes the ReLU activation function.

This property implies that the ReLU activation behaves linearly with respect to the pre-activations along the interpolation path between the two models.

The other condition, reciprocal orthogonality, is defined as follows.

**Definition 5.2** (Reciprocal Orthogonality). We say that two parameters $\boldsymbol{\theta}_a$ and $\boldsymbol{\theta}_b$ satisfy reciprocal orthogonality if, for every hidden layer $\ell \in \{2, 3, \ldots, L\}$, we have $\boldsymbol{z}_{\ell-1}^{(a)} \in \ker \boldsymbol{W}_\ell^{(b)}$ and $\boldsymbol{z}_{\ell-1}^{(b)} \in \ker \boldsymbol{W}_\ell^{(a)}$ almost surely. Equivalently, $\boldsymbol{W}_\ell^{(b)} \boldsymbol{z}_{\ell-1}^{(a)} = 0$ and $\boldsymbol{W}_\ell^{(a)} \boldsymbol{z}_{\ell-1}^{(b)} = 0$.

Reciprocal orthogonality means that multiplying the activations input to layer $\ell$ of one model by the weights at layer $\ell$ of the other model yields zero. When both weak additivity for ReLU activations and reciprocal orthogonality are satisfied, we can derive the following theorem:

**Theorem 5.3.** *For two bias-free models $\boldsymbol{\theta}_a$ and $\boldsymbol{\theta}_b$, if Definitions 5.1 and 5.2 hold, then LEWC is satisfied.*

The proof of Theorem 5.3 is shown in Appendix E.1. Regarding the assumption that biases can be ignored, in models such as ResNet and VGG that employ batch normalization after every convolutional

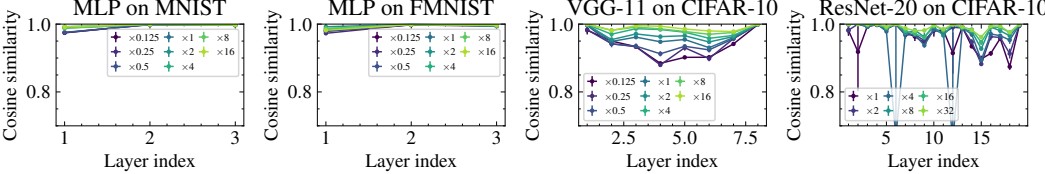

Figure 4: Average cosine similarity between $\sigma((\tilde{z}_\ell^{(a)} + \tilde{z}_\ell^{(b)})/2)$ and $(\sigma(\tilde{z}_\ell^{(a)}) + \sigma(\tilde{z}_\ell^{(b)}))/2$, where $\tilde{z}_\ell^{(a)}$ and $\tilde{z}_\ell^{(b)}$ are the pre-activations of the $\ell$-th layer of two models. Different colors indicate different width expansion factors. The results indicate high cosine similarity for all layers.

layer, the effect of biases can indeed be considered negligible. Moreover, the exponential decay in activation norms is compensated for by the normalization step. Therefore, from Theorem 5.3, LMC can be achieved when Definition 5.1 and Definition 5.2 are satisfied. Thus, in the following two subsections, we empirically confirm whether these two conditions actually hold.

## 5.2 EMPIRICAL VERIFICATION OF WEAK ADDITIVITY FOR RELU ACTIVATIONS

We first verify that ReLU activations behave approximately linearly in sufficiently wide models. Figure 4 shows cosine similarity results used to evaluate ReLU linearity. As seen in the figure, the similarity increases with width, indicating improved linearity.

In the following, we explain from two perspectives why ReLU appears approximately linear when the width is increased. The first reason is the effect of the curse of dimensionality due to the increase in the dimensionality of the intermediate layer outputs. Regarding this point, through an analysis using Gaussian random vectors, we clarify the effect of increasing dimensionality on the ReLU function. The second reason is that when the weights of a trained model become low-rank by widening models, the active neurons of the two models will not overlap. From these two reasons, it is desirable that the width is large and the weights are low-rank to achieve the weak additivity for ReLU activations.

**Curse of dimensionality on ReLU activations.** In high dimensions, two Gaussian random vectors $\boldsymbol{u}$ and $\boldsymbol{v}$ yield high cosine similarity between $\sigma(\boldsymbol{u} + \boldsymbol{v})$ and $\sigma(\boldsymbol{u}) + \sigma(\boldsymbol{v})$ (approximately 0.93).

**Theorem 5.4.** *Let* $\boldsymbol{u}, \boldsymbol{v} \sim \mathcal{N}(\boldsymbol{0}, \boldsymbol{I}_d)$ *be two Gaussian random vectors in* $\mathbb{R}^d$. *Then, with probability at least* $1 - 3\delta$, *for every real number* $\epsilon$ *satisfying* $\epsilon \geq K \max\left(\sqrt{\frac{1}{cd}\log(2/\delta)}, \frac{1}{cd}\log(2/\delta)\right)$,

$$\frac{\frac{3}{4} + \frac{1}{\pi} - \epsilon}{\sqrt{(1 + \epsilon)\left(1 + \frac{1}{\pi} + \epsilon\right)}} \leq \frac{(\sigma(\boldsymbol{u}) + \sigma(\boldsymbol{v}))^\top \sigma(\boldsymbol{u} + \boldsymbol{v})}{\|\sigma(\boldsymbol{u}) + \sigma(\boldsymbol{v})\|\|\sigma(\boldsymbol{u} + \boldsymbol{v})\|} \leq \frac{\frac{3}{4} + \frac{1}{\pi} + \epsilon}{\sqrt{(1 - \epsilon)\left(1 + \frac{1}{\pi} - \epsilon\right)}},$$

*where* $c$ *is a constant and* $K = 32/3$.

As $d$ grows large, $\epsilon \to 0$, so the cosine similarity converges in probability to $(3/4 + 1/\pi)/\sqrt{1 + 1/\pi} \approx 0.93$. Thus, in high dimensions, ReLU behaves almost linearly. While real neural network pre-activations are not Gaussian, this suggests that similar effects arise in practice.

**Low-rank structure reduces overlap among active neurons.** The Gaussian argument implies that a cosine similarity of about 0.93 can be achieved when the dimension is sufficiently large; however, Figure 4 shows cosine similarities higher than 0.93 for wide models, so dimensionality alone does not fully explain the approximate linearity of ReLU. An additional reason is the low-rank structure of weight matrices induced by widening. To illustrate the basic idea, let $\tilde{z}_\ell^{(a)}$ and $\tilde{z}_\ell^{(b)}$ be the pre-activations at some layer $\ell$ in the two models. Let $d_\ell$ be the dimensionality at this layer, and for simplicity assume $d_\ell$ is even. Suppose that in $\tilde{z}_\ell^{(a)}$, all coordinates except the first $d_\ell/2$ are zero (i.e., $\forall i \in \{d_\ell/2+1, d_\ell/2+2, \ldots, d_\ell\}, \tilde{z}_{\ell,i}^{(a)} = 0$), and in $\tilde{z}_\ell^{(b)}$, all coordinates except the last $d_\ell/2$ are zero (i.e., $\forall i \in \{1, 2, \ldots, d_\ell/2, \}, \tilde{z}_{\ell,i}^{(b)} = 0$). Then we can show that $\sigma(\tilde{z}_\ell^{(a)} + \tilde{z}_\ell^{(b)}) = \sigma(\tilde{z}_\ell^{(a)}) + \sigma(\tilde{z}_\ell^{(b)})$ because, for any $i$, $\sigma(\tilde{z}_{\ell,i}^{(a)} + \tilde{z}_{\ell,i}^{(b)}) = \sigma(\tilde{z}_{\ell,i}^{(a)})$ if $i \leq d_\ell/2$, and $\sigma(\tilde{z}_{\ell,i}^{(b)})$ otherwise. On the other hand,

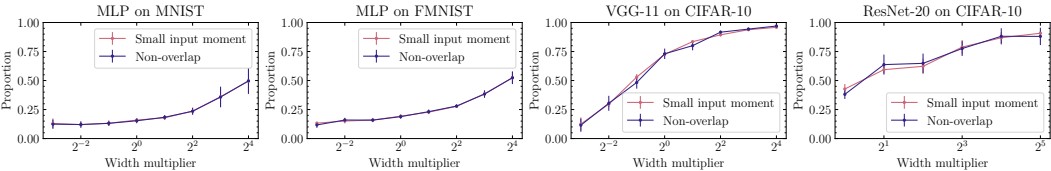

Figure 5: Histogram of the square root of the second (uncentered) moment of ReLU inputs in the second hidden layer (i.e., $\sqrt{\mathbb{E}\tilde{z}_{\ell,i}^2}$). We present results for MLPs with width $\times 16$, a VGG-11 scaled by $\times 16$, and a ResNet-20 scaled by $\times 32$. Most dimensions fall into the leftmost bin, indicating that only a small number of dimensions are active. Results for all layers are provided in Appendix F.7.

Figure 6: Proportion of input dimensions with a small square root of the second moment ("Small input moment") and the proportion of non-overlapping large-moment dimensions ("Non-overlap") between two trained models for the second ReLU layer. Here, "Small input moment" refers to dimensions whose second moment is smaller than one hundredth of the maximum second moment across all dimensions, corresponding to the leftmost bin in Figure 5. As the model width increases, the fraction of small-moment dimensions grows and the overlap decreases.

a similar relation also holds for $\sigma(\tilde{z}_{\ell,i}^{(a)}) + \sigma(\tilde{z}_{\ell,i}^{(b)})$. Thus, in this case, ReLU function behaves linearly.

From the above considerations, for the pre-activations at each layer of the two models, having a smaller overlap among the dimensions whose magnitudes deviate significantly from zero makes the linearity of ReLU more likely to hold. In particular, this is more likely when each weight has low rank. This is because one model's pre-activation is given by $\tilde{z}_\ell^{(a)} = W_\ell^{(a)} z_{\ell-1}^{(a)}$, and if many coordinates of $\tilde{z}_\ell^{(a)}$ are close to zero regardless of the input, then $W_\ell^{(a)}$ must have low rank. If the rank of $W_\ell^{(a)}$ is large, then when regarding $W_\ell^{(a)}$ as a linear mapping, the dimension of its output space is large, so the number of coordinates near zero necessarily decreases. Ito et al. (2025a) showed that as the model width increases, the rank of each layer's weight matrix becomes relatively small compared to the width, suggesting that widening the model induces low-rank weights and makes the linearity of ReLU more feasible.[2]

We empirically verified whether widening the model reduces, for each layer, the relative number of dimensions with a large second moment of the pre-activations. Figure 5 shows the histogram of the square root of the pre-activation second moment at the second hidden layer, demonstrating that most dimensions have negligible second moment. This indicates that only a limited number of dimensions effectively contribute to the model's output. Figure 6 further shows that as the width increases, the overlap between the large-moment dimensions of two independently trained models decreases, supporting the approximate linearity of ReLU.

## 5.3 EMPIRICAL VERIFICATION OF RECIPROCAL ORTHOGONALITY

We next empirically examine whether reciprocal orthogonality holds. Figure 7 reports the ratio $\mathbb{E}\|W_\ell^{(a)} z_{\ell-1}^{(b)}\| / \mathbb{E}\|W_\ell^{(a)} z_{\ell-1}^{(a)}\|$. As model width increases, this ratio decreases across all layers except the input, indicating approximate reciprocal orthogonality. To further confirm this, note that, if re-

---

[2]Low rankness does not by itself imply coordinate level sparsity, which is the more direct driver of weak additivity. Nonetheless, our experiments indicate that the two are tightly coupled in practice: varying weight decay changes the rank of the weight matrices, and we observe a corresponding change in coordinate level sparsity of the pre-activations.

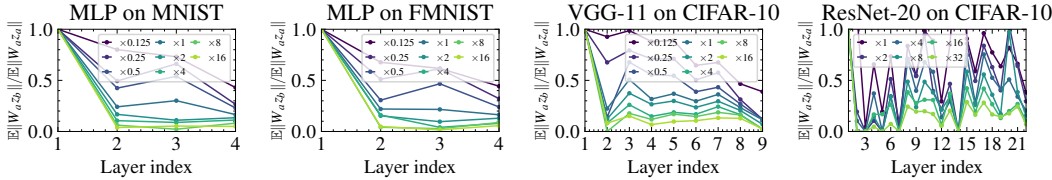

Figure 7: Ratio of mean norms $\mathbb{E}\|\boldsymbol{W}_\ell^{(a)}\boldsymbol{z}_\ell^{(b)}\|/\mathbb{E}\|\boldsymbol{W}_\ell^{(a)}\boldsymbol{z}_\ell^{(a)}\|$ at each layer. The decreasing ratio with width suggests approximate reciprocal orthogonality.

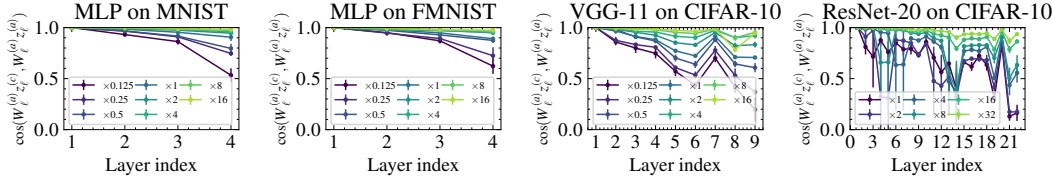

Figure 8: Average cosine similarity between $\boldsymbol{W}_\ell^{(a)}\boldsymbol{z}_{\ell-1}^{(c)}$ and $\boldsymbol{W}_\ell^{(a)}\boldsymbol{z}_{\ell-1}^{(a)}$ for each layer. The similarity increases with model width and approaches one, indicating that reciprocal orthogonality holds.

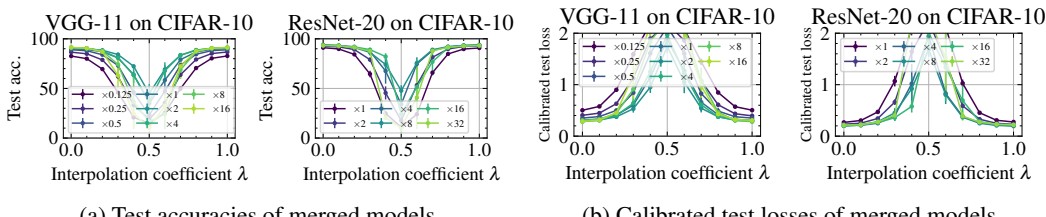

(a) Test accuracies of merged models.      (b) Calibrated test losses of merged models.

Figure 9: Performance of merged models for VGG-11 and ResNet-20 trained with weak weight decay ($10^{-4}$). Note that most of our results in this paper used a weight decay coefficient of 0.003.

ciprocal orthogonality and LEWC hold, then applying the weight matrix of one model (e.g., $\boldsymbol{W}_\ell^{(a)}$) to the merged model's intermediate activation $\boldsymbol{z}_\ell^{(c)}$ should reproduce the corresponding pre-activation of that model (i.e., $\tilde{\boldsymbol{z}}_\ell^{(a)}$). Indeed, $\boldsymbol{W}_\ell^{(a)}\boldsymbol{z}_{\ell-1}^{(c)} = \lambda^\ell \boldsymbol{W}_\ell^{(a)}\boldsymbol{z}_{\ell-1}^{(a)} + (1-\lambda)^\ell \boldsymbol{W}_\ell^{(a)}\boldsymbol{z}_{\ell-1}^{(b)} = \lambda^\ell \boldsymbol{W}_\ell^{(a)}\boldsymbol{z}_{\ell-1}^{(a)}$. To verify this empirically, we compute the average cosine similarity between $\boldsymbol{W}_\ell^{(a)}\boldsymbol{z}_{\ell-1}^{(c)}$ and $\boldsymbol{W}_\ell^{(a)}\boldsymbol{z}_{\ell-1}^{(a)}$ across test data. The results are shown in Figure 8. As expected, the cosine similarity increases with model width and approaches one, confirming that the reciprocal orthogonality holds in practice.

From the above, we confirm that reciprocal orthogonality holds in wide models. This is also attributable to the low-rank structure of the weights. For example, if a trained model's weight matrices have high rank, then since one model's weight $\boldsymbol{W}_\ell^{(a)}$ at some layer $\ell$ has high rank, $\|\boldsymbol{W}_\ell^{(a)}\boldsymbol{z}_{\ell-1}\|/\|\boldsymbol{z}_{\ell-1}\|$ becomes large regardless of the direction of the input $\boldsymbol{z}_{\ell-1}$. This implies that reciprocal orthogonality does not hold. Ito et al. (2025a) clarify that the relative rank of weights with respect to width becomes smaller for wider models, explaining why reciprocal orthogonality is more likely to hold when the model is wider.

## 5.4 EFFECT OF WEIGHT DECAY ON LMC

We argued that the improved performance of wide merged models arises because both weak additivity for ReLU activations and reciprocal orthogonality hold, leading to LEWC. As mentioned, these properties are easier to satisfy when weight matrices are low-rank.

Prior works (Timor et al., 2023; Galanti et al., 2022; Ito et al., 2025a) empirically observed that widening models and applying stronger weight decay encourage low-rank weight matrices. Conversely, when weight decay is weak, weight matrices tend to have higher rank, making LEWC less likely.

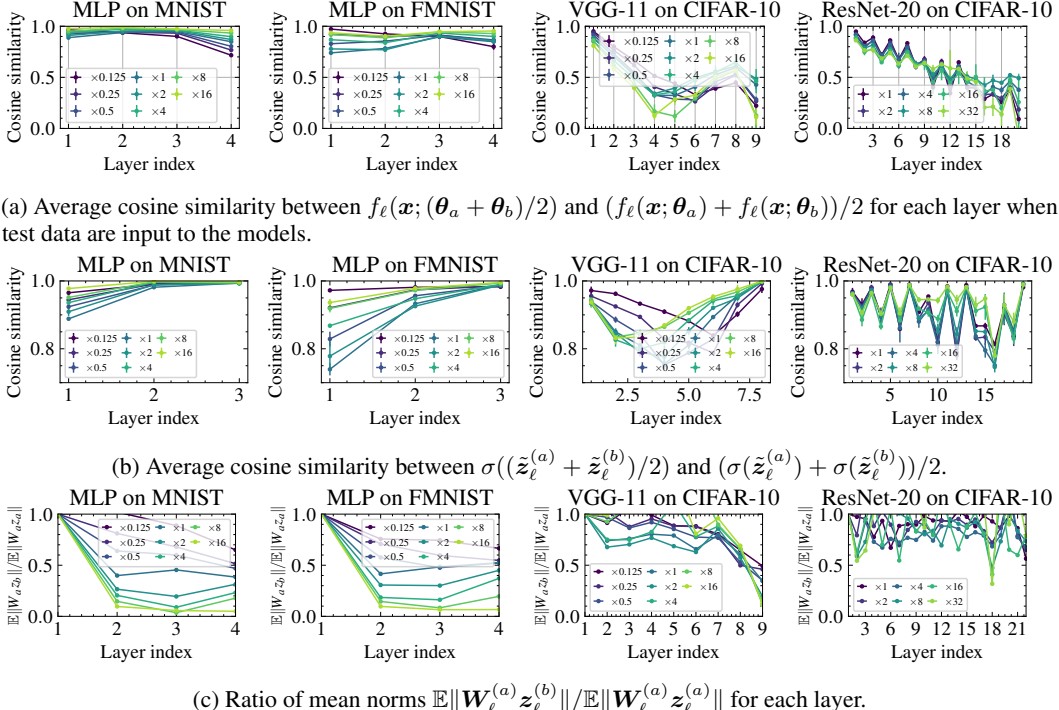

(a) Average cosine similarity between $f_\ell(\boldsymbol{x}; (\boldsymbol{\theta}_a + \boldsymbol{\theta}_b)/2)$ and $(f_\ell(\boldsymbol{x}; \boldsymbol{\theta}_a) + f_\ell(\boldsymbol{x}; \boldsymbol{\theta}_b))/2$ for each layer when test data are input to the models.

(b) Average cosine similarity between $\sigma((\tilde{\boldsymbol{z}}_\ell^{(a)} + \tilde{\boldsymbol{z}}_\ell^{(b)})/2)$ and $(\sigma(\tilde{\boldsymbol{z}}_\ell^{(a)}) + \sigma(\tilde{\boldsymbol{z}}_\ell^{(b)}))/2$.

(c) Ratio of mean norms $\mathbb{E}\|\boldsymbol{W}_\ell^{(a)}\boldsymbol{z}_\ell^{(b)}\|/\mathbb{E}\|\boldsymbol{W}_\ell^{(a)}\boldsymbol{z}_\ell^{(a)}\|$ for each layer.

Figure 10: Experimental results for trained models with weaker weight decay of $10^{-4}$. From top to bottom, the figures show evaluations of LEWC, weak additivity, and reciprocal orthogonality.

Figure 9 shows that merged models trained with weak weight decay indeed exhibit large barriers. To understand this phenomenon in more detail, we examine whether LEWC and its sufficient conditions still hold in this setting. The empirical results are summarized in Figure 10. The impact of weaker weight decay is especially pronounced for VGG-11 and ResNet-20. First, Figure 10(a) indicates that LEWC is no longer satisfied in deeper layers when $\lambda = 1/2$. Moreover, Figures 10(b) and 10(c) show that neither weak additivity of ReLU activations nor reciprocal orthogonality holds. Overall, these results show that weakening weight decay breaks LEWC and its two sufficient conditions, highlighting the role of low-rank structure in enabling LMC.

## 6 CONCLUSION

We empirically showed that simply widening neural networks improves the performance of merged models, eventually matching that of independently trained models. This behavior differs fundamentally from the intuition underlying permutation-based approaches to LMC. We introduced *layerwise exponentially weighted connectivity* (LEWC), which arises when weak additivity for ReLU activations and *reciprocal orthogonality* are satisfied, and we found that these properties hold in sufficiently wide models with low-rank weight matrices. Considering LMC in this distinct setting may provide new insights into both LMC itself and, more broadly, the dynamics of neural network training.

Our experiments focused on standard image classification with relatively simple architectures, since LEWC typically requires larger width multipliers than permutation-based merging. An important direction for future work is to test whether these phenomena extend to larger-scale models and other modalities.

Our work is analytical rather than a proposal of a merging or federated learning method. By identifying weak additivity and reciprocal orthogonality as the drivers of permutation-free LMC, we point to practical directions for developing new model-merging and federated learning methods, such as training or permutation-search procedures that promote these properties, and simple rescaling to offset any LEWC-induced logit norm decay.

ETHICS STATEMENT

This paper presents work whose goal is to advance the field of machine learning. There are many potential societal consequences of our work, none of which we feel must be specifically highlighted here.

REPRODUCIBILITY STATEMENT

The experimental settings for reproducing the results are described in Appendix B. All proofs of the theorems are given in Appendix E.

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

## A   ADDITIONAL RELATED WORK

**(Linear) Mode Connectivity.**   A series of works (Garipov et al., 2018; Draxler et al., 2018; Freeman & Bruna, 2017) have shown that independently trained neural networks can often be connected by low-loss nonlinear paths. Nagarajan & Kolter (2019) provided an early observation that, on MNIST, solutions obtained from stochastic gradient descent (SGD) with the same initialization can be connected by linear paths with little increase in loss. Later, Frankle et al. (2020) systematically demonstrated that such linear connections are not universal: whether they appear depends strongly on the dataset and architecture. They also observed that when two models are branched from a shared partially trained model, the resulting solutions are almost always connected by a linear path. Moreover, they investigated the link between linear mode connectivity (LMC) and the lottery ticket hypothesis (Frankle & Carbin, 2019).

More recent work has explored symmetry and alignment as explanations for LMC. Entezari et al. (2022) conjectured that, once permutation symmetries of hidden units are taken into account, LMC should hold with high probability. Building on this, Ainsworth et al. (2023) introduced a weight-matching approach based on bipartite graph alignment, while Guerrero-Peña et al. (2023) applied Sinkhorn's algorithm as a relaxation to solve the alignment problem more directly. Although a number of papers (Venturi et al., 2019; Nguyen et al., 2019; Nguyen, 2019; Kuditipudi et al., 2019) have studied nonlinear mode connectivity, theoretical understanding of LMC remains limited. Ferbach et al. (2024) established width-dependent conditions under which LMC can be guaranteed, assuming independence of weight vectors. Zhou et al. (2023) proposed the notion of layerwise linear feature connectivity (LLFC) and proved that LLFC implies LMC. More recently, Ito et al. (2025a) highlighted the role of dominant singular vectors in parameter space, showing that they are critical to satisfying LMC, especially when alignment via weight matching is applied.

In addition, several theoretical works have analyzed the effect of network width on nonlinear connectivity. For instance, Nguyen (2019) proved that in over-parameterized neural networks with piecewise-linear activations, every sublevel set of the loss is connected and unbounded. Shevchenko & Mondelli (2020) further showed that in wide multilayer perceptrons, SGD solutions can be linked through piecewise-linear paths along which the loss barrier vanishes as width increases. However, to the best of our knowledge, there are no theoretical results directly establishing that larger width makes LMC more likely. This motivates our empirical investigation of that question.

**Model Merging.**   Model merging has been studied in close connection with LMC, as well as in the context of federated learning. The idea of federated learning was introduced by McMahan et al. (2017) and Konečný et al. (2016), where models are trained locally on partitioned datasets. Wang et al. (2020) explored permuting local model components prior to aggregation, while Singh & Jaggi (2020) proposed an approach to merge models by optimal transport, conceptually related to the weight-matching method of Ainsworth et al. (2023). Although the method of Singh et al. (2024) was primarily designed for fusion and empirically underperforms weight matching, it can still be interpreted as an LMC-based technique because it enforces alignment within the same architecture.

While our work focuses on merging models trained from different random seeds, there is a parallel line of research on merging models obtained by fine-tuning from the same fundamental model. A representative example is Model Soups (Wortsman et al., 2022), which showed that averaging weights across fine-tuned models trained under different hyperparameters can improve test accuracy without additional inference cost, contrasting with standard ensembling. Matena & Raffel (2022) extended this idea by weighting models according to the Fisher information matrix, leading to more effective combinations. Yadav et al. (2023) introduced TIES-Merging, which prunes parameters with small updates, resolves conflicting signs, and averages only aligned components. These approaches are primarily designed for merging fine-tuned models of the same pretrained model, not for models independently trained from scratch with different initializations. By contrast, our study addresses this more challenging setting and seeks to shed light on the feasibility of model merging under such conditions.

**Relation between Model Width and Loss Landscape.**   A broad trend in prior work is that increasing width tends to simplify the loss landscape of deep neural networks and makes the set of optimal solutions more connected. Nguyen (2021); Simsek et al. (2021) pointed out that the topology of the optimal solution set can change drastically by adding only a small number of neurons.

Specifically, Nguyen (2021) proved that for deep networks, having a single wide layer of width $n + 1$, where $n$ is the number of training samples, is sufficient to ensure that all sublevel sets of the training loss are connected. Simsek et al. (2021) further showed that adding one extra neuron per layer is enough to turn discrete minima into a single connected manifold. Liang et al. (2018) showed that adding one special neuron to each layer can eliminate all bad local minima, so that every local minimum becomes a global minimum. Subsequent works have also reported phase-transition-like behavior as width increases. For example, Li et al. (2022) showed that there exists a critical width $m^*$ such that when $m \geq m^*$, all suboptimal basins disappear and only optimal basins remain. More recently, Kim et al. (2025) established that in regularized two-layer networks, once the width exceeds a certain threshold, the optimal solution set becomes connected in parameter space. These results collectively suggest that widening changes the structure of the optimal set and can eventually yield mode connectivity, especially under regularization. Building on this line of work, we empirically show that, after calibrating the loss via inverse temperature scaling, the stronger property of linear mode connectivity holds in sufficiently wide networks without permutations. We believe our findings point to new directions for understanding SGD learning dynamics and may also inform future techniques for model merging and federated learning.

## B  EXPERIMENTAL SETTINGS

This section details the setup used to train neural networks and obtain solutions from stochastic gradient descent (SGD). The experiments were conducted on four benchmark datasets: MNIST (Lecun et al., 1998), Fashion-MNIST (FMNIST) (Xiao et al., 2017), CIFAR-10 (Krizhevsky & Hinton, 2009), and CIFAR-100. For weight matching (WM), we adopted the implementation available in the public repository of Ainsworth et al. (2023)[3].

All training and evaluation procedures were performed on a Linux workstation equipped with two AMD EPYC 7543 32-core processors, eight NVIDIA A30 GPUs, and 512 GB of memory. We used PyTorch 2.5.1[4], PyTorch Lightning 2.4.0[5], and torchvision 0.20.1[6] as the software framework.

### B.1  MODEL TRAINING

**MLP on MNIST and FMNIST.**   In line with the setup of Ainsworth et al. (2023), we trained a fully connected multilayer perceptron (MLP) with three hidden layers, each containing 512 units. ReLU activations were applied to the hidden layers. For both MNIST and FMNIST, training used the Adam optimizer with a fixed learning rate of $1 \times 10^{-3}$ and a weight decay of $3 \times 10^{-3}$. The batch size was 512, and models were trained for up to 100 epochs. No learning rate scheduling was applied.

**VGG-11 and ResNet-20 on CIFAR-10 and CIFAR-100.**   For these experiments, we employed the source code released by Ito et al. (2025b)[7]. The VGG-11 and ResNet-20 architectures followed the implementations described by Ainsworth et al. (2023). During model merging, BatchNorm statistics were re-calibrated using the training data, following the procedure of Jordan et al. (2023). Models were optimized with SGD using a learning rate of $0.4$ and a weight decay of $3 \times 10^{-3}$. The batch size was set to 500, and training was carried out for a maximum of 100 epochs. Data augmentation included random $32 \times 32$ crops and random horizontal flips.

## C  LAYERWISE LINEAR FEATURE CONNECTIVITY

Zhou et al. (2023) pointed out that the existence of LMC in settings such as permutation and spawning can be attributed to a property called layerwise linear feature connectivity (LLFC). Additionally, Zhou et al. (2023) also showed that two conditions are sufficient for LLFC to hold: weak additivity for ReLU activations (Definition 5.1) and the commutativity property defined as follows:

---

[3] https://github.com/samuela/git-re-basin
[4] https://pytorch.org/
[5] https://lightning.ai/docs/pytorch/stable/
[6] https://pytorch.org/vision/stable/index.html
[7] https://github.com/e5-a/STE-MM

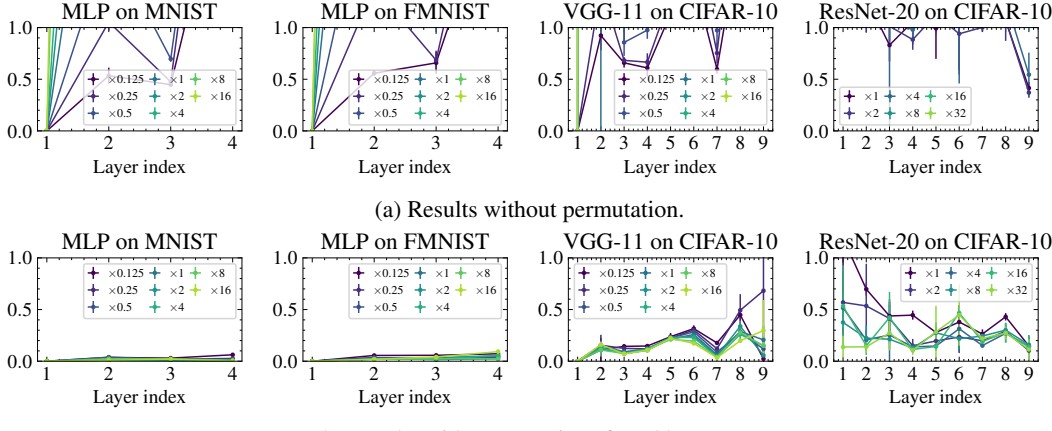

(a) Results without permutation.

(b) Results with permutations found by WM.

Figure 11: Verification of whether Definition C.1 holds at each layer. For all datasets, we compute $\text{dist}(\text{vec}(\boldsymbol{W}_\ell^{(a)} \boldsymbol{z}_{\ell-1}^{(a)} + \boldsymbol{W}_\ell^{(b)} \boldsymbol{z}_{\ell-1}^{(b)}), \text{vec}(\boldsymbol{W}_\ell^{(a)} \boldsymbol{z}_{\ell-1}^{(b)} + \boldsymbol{W}_\ell^{(b)} \boldsymbol{z}_{\ell-1}^{(a)}))$ at each layer. A smaller value indicates a higher degree of commutativity. For reference, we also report the results using permutations found by WM in Figure 11(b). Without permutations, the values are very large for all layers except the input layer, implying that the commutativity property almost never holds.

**Definition C.1** (Commutativity). Let $\boldsymbol{W}_\ell^{(a)}$ and $\boldsymbol{W}_\ell^{(b)}$ be the weights, and $\boldsymbol{z}_\ell^{(a)}$ and $\boldsymbol{z}_\ell^{(b)}$ the outputs of the $\ell$-th layer. The models $\boldsymbol{\theta}_a$ and $\boldsymbol{\theta}_b$ satisfy *commutativity* if, for every layer $\ell \in [L]$,

$$\boldsymbol{W}_\ell^{(a)} \boldsymbol{z}_{\ell-1}^{(a)} + \boldsymbol{W}_\ell^{(b)} \boldsymbol{z}_{\ell-1}^{(b)} = \boldsymbol{W}_\ell^{(a)} \boldsymbol{z}_{\ell-1}^{(b)} + \boldsymbol{W}_\ell^{(b)} \boldsymbol{z}_{\ell-1}^{(a)} \quad \text{almost surely.} \tag{2}$$

Since this condition can be rewritten as $(\boldsymbol{W}_\ell^{(a)} - \boldsymbol{W}_\ell^{(b)})(\boldsymbol{z}_{\ell-1}^{(a)} - \boldsymbol{z}_{\ell-1}^{(b)}) = 0$, it tends to hold when the weights of the two models are close (i.e., $\|\boldsymbol{W}_\ell^{(a)} - \boldsymbol{W}_\ell^{(b)}\| \approx 0$). This is precisely the objective of WM methods, which explains why WM encourages the emergence of LLFC.

We confirmed in Section 4 that LEWC holds when $\lambda = 1/2$. However, we cannot rule out the possibility that LLFC also holds. It is possible that the cosine similarity in Figure 3 approaching 1 is due to the validity of LLFC. To exclude the possibility, we confirm that the commutativity property, which is one of the sufficient conditions for LLFC, does not hold in our settings. Figure 11(a) presents experimental results on whether the commutativity property holds across layers for all models. For comparison, we also show the results when applying the permutations discovered by WM in Figure 11(b). Following prior work, we evaluate the difference between the left- and right-hand sides of Equation (2) using $\text{dist}(x, y) = \|x - y\|^2 / (\|x\| \|y\|)$. For ResNet-20, we compute this only for the first convolutional layer of each block. As the results show, without permutations, the commutativity property hardly holds, indicating that LLFC is unlikely to be satisfied in our settings.

## C.1 DIFFERENCE BETWEEN LLFC AND LEWC

We empirically confirmed that, in LMC without permutations, LLFC does not hold whereas LEWC does. In this subsection, we provide a qualitative explanation for why LLFC fails even though LEWC holds.

A key difference between LLFC and LEWC is that LLFC is a property that arises when the weights of two models are sufficiently close, whereas LEWC is a property that arises when the weights of two models are different, namely approximately orthogonal. This can be understood by focusing on the difference between commutativity and reciprocal orthogonality, which are the respective sufficient conditions that distinguish LLFC from LEWC.

Below we explain this point more concretely using equations. Let be $\boldsymbol{\theta}_a$ and $\boldsymbol{\theta}_b$ the two models, and for simplicity assume that biases can be ignored. Commutativity for $\ell \geq 1$ requires

$$\boldsymbol{W}_\ell^{(a)} \boldsymbol{z}_{\ell-1}^{(a)} + \boldsymbol{W}_\ell^{(b)} \boldsymbol{z}_{\ell-1}^{(b)} = \boldsymbol{W}_\ell^{(a)} \boldsymbol{z}_{\ell-1}^{(b)} + \boldsymbol{W}_\ell^{(b)} \boldsymbol{z}_{\ell-1}^{(a)} \Leftrightarrow (\boldsymbol{W}_\ell^{(a)} - \boldsymbol{W}_\ell^{(b)})(\boldsymbol{z}_{\ell-1}^{(a)} - \boldsymbol{z}_{\ell-1}^{(b)}) = 0.$$

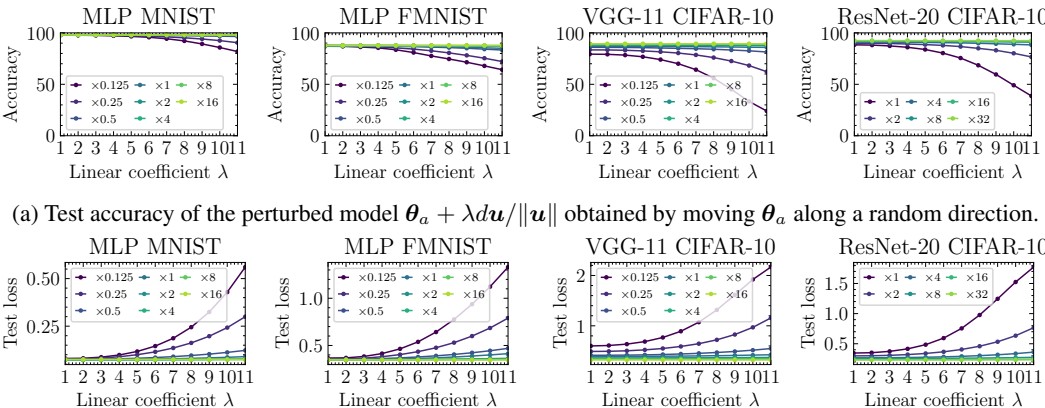

(a) Test accuracy of the perturbed model $\boldsymbol{\theta}_a + \lambda d\boldsymbol{u}/\|\boldsymbol{u}\|$ obtained by moving $\boldsymbol{\theta}_a$ along a random direction.

(b) Uncalibrated test loss of the perturbed model $\boldsymbol{\theta}_a + \lambda d\boldsymbol{u}/\|\boldsymbol{u}\|$ obtained by moving $\boldsymbol{\theta}_a$ along a random direction.

Figure 12: Test accuracy and test loss of models perturbed along a random direction.

This condition clearly holds when all weights match, that is, when $\|\boldsymbol{W}_\ell^{(a)} - \boldsymbol{W}_\ell^{(b)}\| = 0$. Zhou et al. (2023) stated that this explains why WM can find permutations that make LMC hold, since the objective of WM is to search for permutations that minimize the distance between the weights of two models. In spawning, it is also known that LMC holds between solutions obtained by branching from a partially trained initialization and then continuing SGD independently. This is because the first training steps before branching keep the two models close, which promotes commutativity. Zhou et al. (2023) experimentally verified that commutativity holds in the spawning setting as well. In contrast, reciprocal orthogonality clearly does not hold when the two weight matrices match. Specifically, it requires $\boldsymbol{W}_\ell^{(a)}\boldsymbol{z}_\ell^{(b)} = 0$ and $\boldsymbol{W}_\ell^{(b)}\boldsymbol{z}_\ell^{(a)} = 0$, but if the weight matrices are equal and biases are negligible, then the activations are also equal, and these conditions are not satisfied. This shows that commutativity requires the two models to be sufficiently close, whereas reciprocal orthogonality requires them to be sufficiently different, namely approximately orthogonal, so the two requirements are incompatible. For example, in Figures 7 and 11(a) we showed that without permutations, commutativity fails while reciprocal orthogonality holds, which reflects that the two models remain sufficiently distinct. Conversely, when models are aligned using WM discovered permutations, or in the spawning setting, the weight matrices are expected to be close. In such cases, commutativity is likely to hold, whereas reciprocal orthogonality is unlikely to be satisfied.

This incompatibility implies that the reason for LMC identified in our paper is fundamentally different from the LLFC based explanation of LMC in Zhou et al. (2023). Intuitively, under commutativity, permutations make the two models sufficiently close, so the merged model also remains close to each of the original models, and its output stays similar to theirs. In contrast, in our explanation, the two models have weight matrices that are highly different, namely approximately orthogonal, which allows the merged model to embed the two functions independently.

This distinction also explains why inverse temperature calibration is necessary under LEWC. LLFC requires the two weight matrices to be close, so the norm of the merged weight matrix is similar to that of the original models. In contrast, LEWC requires the two weight matrices to be orthogonal, so the norm of the merged weight matrix is strictly smaller than that of each original model. Since this reduction occurs at every layer, the output norm decays exponentially with depth. This provides the reason for the exponential decay of activation norms under LEWC.

# D RELATION TO LOSS LANDSCAPE FLATNESS

We showed that LMC becomes easier to satisfy as the width increases. One possible alternative explanation is that widening simply improves the flatness of the loss or accuracy landscape, and that

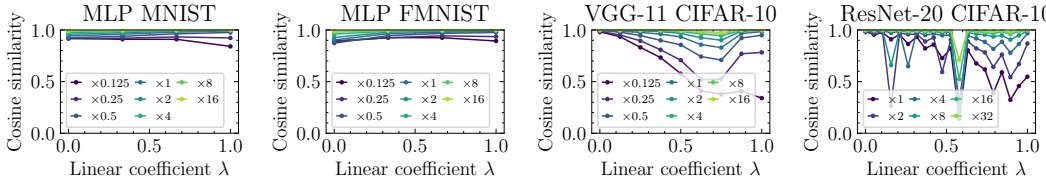

Figure 13: Cosine similarity of hidden layer activations between $\boldsymbol{\theta}_a$ and the perturbed model $\boldsymbol{\theta}_a + d\boldsymbol{u}/\|\boldsymbol{u}\|$.

LMC emerges as a consequence of this increased flatness. In this section, we examine this possibility and show that a mere improvement in flatness is insufficient to explain our observations.

## D.1 EMPIRICAL EVALUATION OF FLATNESS

As a simple proxy for flatness with respect to width, we evaluate how test accuracy and test loss change when we perturb a trained model along a random direction. Let $\boldsymbol{\theta}_a$ and $\boldsymbol{\theta}_b$ be two trained models. We first compute their $L^2$ distance $d = \|\boldsymbol{\theta}_a - \boldsymbol{\theta}_b\|$. Although $d$ could in principle grow with width, in our setting weight decay prevents it from increasing substantially. We then generate a Gaussian random vector $\boldsymbol{u}$ and normalize it to unit length, and use a perturbation of the same norm as the inter model distance, namely $d\boldsymbol{u}/\|\boldsymbol{u}\|$. This construction is intended to compare random perturbations at the same scale as the distance between two independently trained models. For each $\lambda \in [0, 1]$, we measure the accuracy and loss of the perturbed model $\boldsymbol{\theta}_a + \lambda d\boldsymbol{u}/\|\boldsymbol{u}\|$. We expect performance to degrade as $\lambda$ approaches 1, and we interpret the rate of degradation as a proxy for flatness. The results are shown in Figure 12. As the width increases, both test accuracy and test loss become almost insensitive to the perturbation. In particular, even at $\lambda = 1$, the degradation becomes negligible after a modest increase in width. This indicates that this flatness proxy improves rapidly with width.

## D.2 WHY DOES THE FLATNESS IMPROVE WITH WIDTH

Intuitively, one might expect that moving model parameters in a random direction would quickly harm accuracy and loss. However, Figure 12 shows that widening significantly suppresses this degradation, even more so than in the interpolation between two independently trained models shown in Figures 1 and 2. We attribute this to the nature of the random perturbation. When the width is large, a perturbation with fixed overall norm is spread over a much larger number of parameters, so the per parameter magnitude becomes relatively small. As a result, the perturbed model behaves similarly to the original one.

More formally, let $\boldsymbol{W}_\ell$ denote the weight matrix of the original model at layer $\ell$, and let $\boldsymbol{W}_\ell'$ denote the random perturbation added to that layer. The perturbed weights are $\boldsymbol{W}_\ell + \boldsymbol{W}_\ell'$. Ignoring biases, the pre-activation becomes $(\boldsymbol{W}_\ell + \boldsymbol{W}_\ell')\boldsymbol{z}_{\ell-1} = \boldsymbol{W}_\ell \boldsymbol{z}_{\ell-1} + \boldsymbol{W}_\ell' \boldsymbol{z}_{\ell-1}$. Thus, the change in output induced by the perturbation is represented by $\boldsymbol{W}_\ell' \boldsymbol{z}_{\ell-1}$. The $i$ th component of this term is $\sum_j \boldsymbol{W}_{\ell,i,j}' \boldsymbol{z}_{\ell-1,j}$. Since $\boldsymbol{W}_\ell'$ is independent of $\boldsymbol{z}_{\ell-1}$ and its total norm is fixed to $d$, the typical magnitude of each entry $\boldsymbol{W}_{\ell,i,j}'$ decreases as the width increases, because the same norm is distributed across more parameters. Moreover, due to the effect of weight decay, the inter model distance $d$ does not necessarily grow with width. Therefore, $\sum_j \boldsymbol{W}_{\ell,i,j}' \boldsymbol{z}_{\ell-1,j}$ is expected to approach zero as the width increases.

To validate this intuition, we measure the cosine similarity of hidden layer activations between the original model $\boldsymbol{\theta}_a$ and the perturbed model $\boldsymbol{\theta}_a + d\boldsymbol{u}/\|\boldsymbol{u}\|$. The results are shown in Figure 13. The similarity increases with width, supporting the view that the apparent improvement in flatness under random perturbations is largely explained by the diminishing effect of Gaussian noise in wide networks.

## D.3 FLATNESS ALONE CANNOT EXPLAIN OUR FINDINGS

As a comparison to the random perturbation setting, we examine the cosine similarity of hidden layer activations between two independently trained models $\boldsymbol{\theta}_a$ and $\boldsymbol{\theta}_b$, shown in Figure 14. Even though

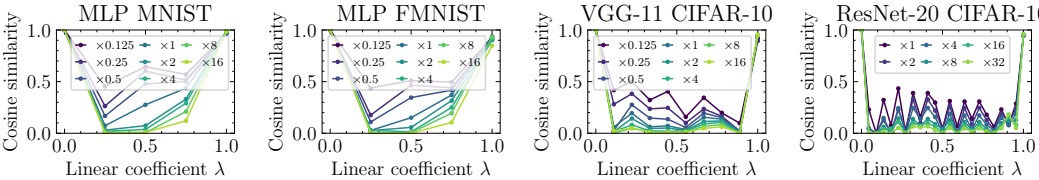

Figure 14: Cosine similarity of hidden layer activations between $\boldsymbol{\theta}_a$ and $\boldsymbol{\theta}_b$.

the norm of the random perturbation in Figure 13 equals the model distance between $\boldsymbol{\theta}_a$ and $\boldsymbol{\theta}_b$, the activation similarities exhibit qualitatively different behavior. This suggests that probing flatness via random perturbations does not capture the relationship between independently trained models produced by SGD.

Finally, note that if LMC were explained solely by increased flatness, then the exponential decay of activation norms predicted by LEWC would also need to follow from flatness, which it does not. Moreover, if flatness were the only reason, the merged model should achieve test loss comparable to the original models even without inverse temperature calibration. However, Figure 2 contradicts this. These observations indicate that improved flatness alone is insufficient to account for the emergence of LMC with increasing width.

## E   PROOF

### E.1   MAIN THEOREM

**Theorem E.1.** *For two bias-free models $\boldsymbol{\theta}_a$ and $\boldsymbol{\theta}_b$, if Definition 5.1 and Definition 5.2 hold, then LEWC is satisfied.*

*Proof.* We prove the claim by induction on the depth $\ell$. For the base case $\ell = 1$, the statement holds trivially. Now assume $\ell \geq 2$ and that Equation (1) holds for all layers prior to $\ell$. The left-hand side of Equation (1) can be written as

$$
\begin{aligned}
f_\ell(\boldsymbol{x}; \lambda\boldsymbol{\theta}_a + (1-\lambda)\boldsymbol{\theta}_b) &= \sigma\Big(\big(\lambda\boldsymbol{W}_\ell^{(a)} + (1-\lambda)\boldsymbol{W}_\ell^{(b)}\big) f_{\ell-1}(\boldsymbol{x}; \lambda\boldsymbol{\theta}_a + (1-\lambda)\boldsymbol{\theta}_b)\Big) \\
&= \sigma\Big(\big(\lambda\boldsymbol{W}_\ell^{(a)} + (1-\lambda)\boldsymbol{W}_\ell^{(b)}\big) \big(\lambda^{\ell-1} f_{\ell-1}(\boldsymbol{x}; \boldsymbol{\theta}_a) + (1-\lambda)^{\ell-1} f_{\ell-1}(\boldsymbol{x}; \boldsymbol{\theta}_b)\big)\Big) \\
&= \lambda^\ell \sigma(\boldsymbol{W}_\ell^{(a)} f_{\ell-1}(\boldsymbol{x}; \boldsymbol{\theta}_a)) + (1-\lambda)^\ell \sigma(\boldsymbol{W}_\ell^{(b)} f_{\ell-1}(\boldsymbol{x}; \boldsymbol{\theta}_b)) \\
&= \lambda^\ell f_\ell(\boldsymbol{x}; \boldsymbol{\theta}_a) + (1-\lambda)^\ell f_\ell(\boldsymbol{x}; \boldsymbol{\theta}_b).
\end{aligned}
$$

Thus, the statement holds for layer $\ell$, completing the induction. $\square$

### E.2   PROOF OF THEOREM 5.4

We now provide the proof of the following theorem.

**Theorem E.2.** *Let $\boldsymbol{u}, \boldsymbol{v} \sim \mathcal{N}(\boldsymbol{0}, \boldsymbol{I}_d)$ be two independent Gaussian random vectors in $\mathbb{R}^d$. Then, with probability at least $1 - 3\delta$, for every real number $\epsilon$ satisfying $\epsilon \geq K \max\left(\sqrt{\frac{1}{cd} \log(2/\delta)}, \frac{1}{cd} \log(2/\delta)\right)$, we have*

$$
\frac{\frac{3}{4} + \frac{1}{\pi} - \epsilon}{\sqrt{(1+\epsilon)\left(1 + \frac{1}{\pi} + \epsilon\right)}} \leq \frac{(\sigma(\boldsymbol{u}) + \sigma(\boldsymbol{v}))^\top \sigma(\boldsymbol{u} + \boldsymbol{v})}{\|\sigma(\boldsymbol{u}) + \sigma(\boldsymbol{v})\|\|\sigma(\boldsymbol{u} + \boldsymbol{v})\|} \leq \frac{\frac{3}{4} + \frac{1}{\pi} + \epsilon}{\sqrt{(1-\epsilon)\left(1 + \frac{1}{\pi} - \epsilon\right)}},
$$

*where $c$ is an absolute constant and $K = 32/3$.*

To this end, we prepare several auxiliary results. First, we recall the definitions of sub-Gaussian and sub-exponential random variables.

**Definition E.3** (Sub-Gaussian random variable ([Vershynin, 2018](#))). A random variable $X$ is called *sub-Gaussian* if there exists a constant $C > 0$ such that, for all $t \geq 0$,

$$\Pr(|X| \geq t) \ \leq \ 2\exp(-t^2/C^2).$$

The sub-Gaussian norm is defined as

$$\|X\|_{\psi_2} := \inf\left\{c > 0 : \mathbb{E}\exp(X^2/c^2) \leq 2\right\}.$$

**Definition E.4** (Sub-exponential random variable ([Vershynin, 2018](#))). A random variable $X$ is called *sub-exponential* if there exists a constant $C > 0$ such that, for all $t \geq 0$,

$$\Pr(|X| \geq t) \ \leq \ 2\exp(-t/C).$$

The sub-exponential norm is defined as

$$\|X\|_{\psi_1} := \inf\left\{t > 0 : \mathbb{E}\exp(|X|/t) \leq 2\right\}.$$

It is standard that if $X$ is sub-Gaussian, then $\|X^2\|_{\psi_1} = \|X\|_{\psi_2}^2$. Moreover, for two sub-Gaussian random variables $X$ and $Y$, we have $\|XY\|_{\psi_1} \leq \|X\|_{\psi_2}\|Y\|_{\psi_2}$.

We also need the following classical identity for correlated Gaussians.

**Theorem E.5.** *Let $(x, y)$ be jointly Gaussian random variables with zero mean, unit variances, and correlation $\rho$. Then*

$$\mathbb{E}\big[\sigma(x)\sigma(y)\big] = \frac{1}{4}\left(\rho + \frac{2}{\pi}\left(\sqrt{1 - \rho^2} + \rho\arcsin\rho\right)\right).$$

*Proof.* Note that $\sigma(x) = (x + |x|)/2$. Expanding gives

$$\mathbb{E}[\sigma(x)\sigma(y)] = \tfrac{1}{4}\mathbb{E}[xy + x|y| + |x|y + |x||y|].$$

We have $\mathbb{E}[xy] = \rho$. For the cross terms, using $\mathbb{E}[x \mid y] = \rho y$, we obtain

$$\mathbb{E}[x|y|] = \mathbb{E}[|y|\,\mathbb{E}[x \mid y]] = \rho\mathbb{E}[y|y|] = 0,$$

since $y \mapsto y|y|$ is an odd function under the symmetric Gaussian distribution. Similarly $\mathbb{E}[|x|y] = 0$. Hence,

$$\mathbb{E}[\sigma(x)\sigma(y)] = \tfrac{1}{4}\big(\rho + \mathbb{E}|x||y|\big).$$

It is a classical fact on absolute moments of Gaussians ([Haas, 2018](#)) that

$$\mathbb{E}|x||y| = \frac{2}{\pi}\left(\sqrt{1 - \rho^2} + \rho\arcsin\rho\right),$$

which completes the proof. $\square$

*Proof of Theorem 5.4.* Writing the cosine similarity elementwise, we have

$$\frac{(\sigma(\boldsymbol{u}) + \sigma(\boldsymbol{v}))^\top \sigma(\boldsymbol{u} + \boldsymbol{v})}{\|\sigma(\boldsymbol{u}) + \sigma(\boldsymbol{v})\|\,\|\sigma(\boldsymbol{u} + \boldsymbol{v})\|} = \frac{\sum_i (\sigma(u_i) + \sigma(v_i))\sigma(u_i + v_i)}{\sqrt{(\sum_i \sigma(u_i + v_i)^2)(\sum_i (\sigma(u_i) + \sigma(v_i))^2)}}.$$

By Theorem E.5, straightforward calculations yield

$$\mathbb{E}[(\sigma(u_i) + \sigma(v_i))\sigma(u_i + v_i)] = \tfrac{3}{4} + \tfrac{1}{\pi}, \quad \mathbb{E}[\sigma(u_i + v_i)^2] = 1, \quad \mathbb{E}[(\sigma(u_i) + \sigma(v_i))^2] = 1 + \tfrac{1}{\pi}.$$

Moreover, we can bound the sub-exponential norms:

$$\|(\sigma(u_i) + \sigma(v_i))\sigma(u_i + v_i)\|_{\psi_1} \ \leq \ \|\sigma(u_i) + \sigma(v_i)\|_{\psi_2}\|\sigma(u_i + v_i)\|_{\psi_2} \ \leq \ 2\|\sigma(u_i)\|_{\psi_2}\|u_i + v_i\|_{\psi_2} \ \leq \ \tfrac{32}{3}.$$

Similarly, $\|\sigma(u_i + v_i)^2\|_{\psi_1} \leq 32/3$ and $\|(\sigma(u_i) + \sigma(v_i))^2\|_{\psi_1} \leq 32/3$. Hence each of these random variables is sub-exponential. Applying Bernstein's inequality ([Vershynin, 2018](#)) with $K = 32/3$, we obtain

$$\Pr\left(\left|\frac{1}{d}\sum_i (\sigma(u_i) + \sigma(v_i))\sigma(u_i + v_i) - \tfrac{3}{4} - \tfrac{1}{\pi}\right| \geq t\right) \leq 2\exp\left(-cd\min\left(\tfrac{t^2}{K^2}, \tfrac{t}{K}\right)\right), \quad (3)$$

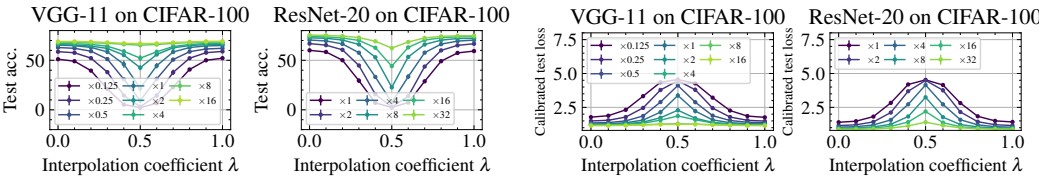

(a) Test accuracies of merged models.          (b) Calibrated test losses of merged models.

Figure 15: Performance of merged models for VGG-11 and ResNet-20 trained on the CIFAR-100 dataset.

$$\Pr\left(\left|\frac{1}{d}\sum_i (\sigma(u_i) + \sigma(v_i))^2 - (1 + 1/\pi)\right| \geq t\right) \leq 2\exp\left(-cd\min\left(\frac{t^2}{K}, \frac{t}{K}\right)\right), \tag{4}$$

$$\Pr\left(\left|\frac{1}{d}\sum_i \sigma(u_i + v_i)^2 - 1\right| \geq t\right) \leq 2\exp\left(-cd\min\left(\frac{t^2}{K}, \frac{t}{K}\right)\right), \tag{5}$$

where $c$ is a constant.

Now set $\delta \geq 2\exp(-cd\min(t^2/K^2, t/K))$. Equivalently,

$$t \geq K\max\left(\sqrt{\frac{1}{cd}\log(2/\delta)}, \frac{1}{cd}\log(2/\delta)\right).$$

Letting $\epsilon$ denote this bound, equation 3 implies that with probability at least $1 - \delta$,

$$\tfrac{3}{4} + \tfrac{1}{\pi} - \epsilon \leq \tfrac{1}{d}\sum_i (\sigma(u_i) + \sigma(v_i))\sigma(u_i + v_i) \leq \tfrac{3}{4} + \tfrac{1}{\pi} + \epsilon.$$

Analogous statements hold for equation 4 and equation 5. Combining these inequalities yields

$$\frac{\tfrac{3}{4} + \tfrac{1}{\pi} - \epsilon}{\sqrt{(1+\epsilon)(1 + 1/\pi + \epsilon)}} \leq \frac{\sum_i (\sigma(u_i) + \sigma(v_i))\sigma(u_i + v_i)}{\sqrt{(\sum_i \sigma(u_i + v_i)^2)(\sum_i (\sigma(u_i) + \sigma(v_i))^2)}} \leq \frac{\tfrac{3}{4} + \tfrac{1}{\pi} + \epsilon}{\sqrt{(1-\epsilon)(1 + 1/\pi - \epsilon)}}$$

with probability at least $1 - 3\delta$, completing the proof. □

## F  ADDITIONAL EXPERIMENTAL RESULTS

### F.1  CIFAR-100 DATASET

In this work, we primarily focused on experiments with relatively simple datasets such as MNIST and CIFAR-10. This choice is due to the fact that LMC does not hold unless the models are made sufficiently wide, which makes verification difficult on more complex datasets. In this section, we present experiments conducted on CIFAR-100, a more challenging dataset, to investigate whether LMC can also be achieved in this setting. Figure 15 reports the accuracy and calibrated test loss when changing the merging ratio $\lambda$. We observe that increasing the model width leads to a monotonic decrease of the barrier in both metrics, eventually eliminating it. Notably, while previous permutation-based merging methods required that the original models achieve sufficiently high test accuracy and low test loss in order for LMC to hold, our permutation-free approach demonstrates that LMC can be achieved even when the models' performance is comparatively modest.

### F.2  PERMUTATION-BASED METHOD

The test accuracy values obtained by merging with the permutations found by WM are shown in Figure 16. While the use of permutations improves performance compared to the case without permutations (Figure 1), the gap nearly vanishes as the model width becomes sufficiently large.

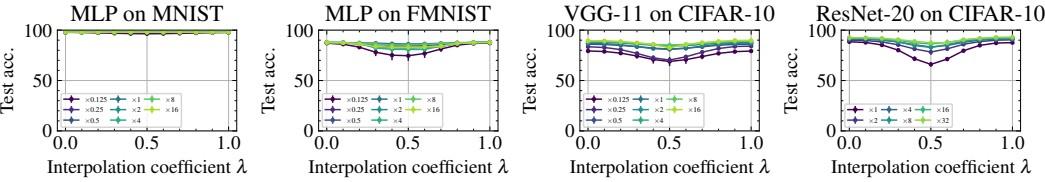

Figure 16: Test accuracies of merged models **with permutations** found by WM.

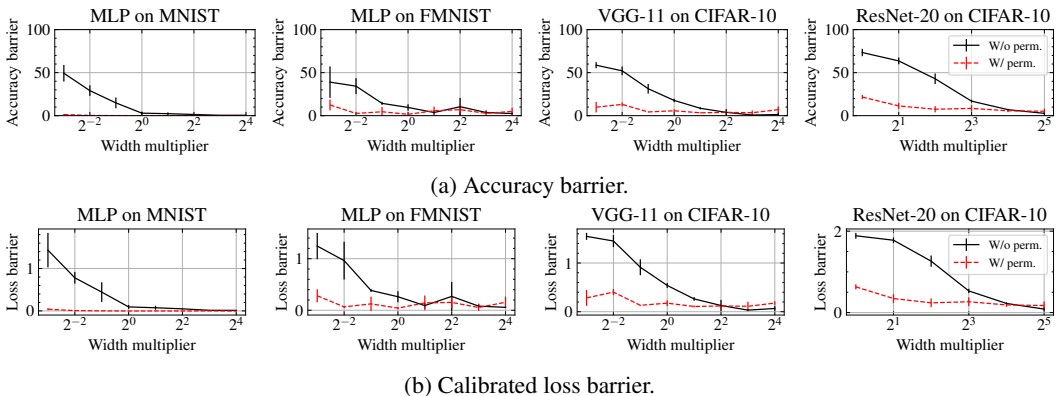

(a) Accuracy barrier.

(b) Calibrated loss barrier.

Figure 17: Accuracy and calibrated loss barriers with and without the permutations identified by WM.

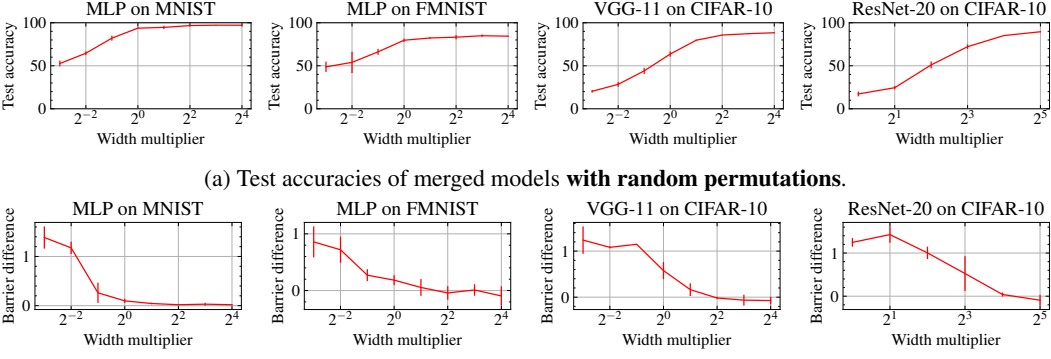

(a) Test accuracies of merged models **with random permutations**.

(b) Difference in calibrated loss barriers between random permutations and the permutation identified by WM.

Figure 18: Effects of random permutations on accuracy and loss barriers.

### F.3 LOSS AND ACCURACY BARRIERS

Figure 17 shows how the barrier changes with increasing width with and without the permutation found by WM. We observe that applying the permutation tends to reduce the barrier regardless of the width. However, when the width is sufficiently large, the barrier becomes nearly zero regardless of whether the permutation is used.

### F.4 RANDOM PERMUTATIONS

For two models $\theta_a$ and $\theta_b$ trained by SGD, we applied randomly chosen permutations $\pi_a$ and $\pi_b$ to each model, and merged the resulting models $\pi_a(\theta_a)$ and $\pi_b(\theta_b)$. The corresponding test accuracies are shown in Figure 18(a). We observe that, when the width is sufficiently large, high test accuracy is obtained regardless of the choice of permutations. Furthermore, Figure 18(b) shows the difference in calibrated loss barriers between using random permutations and using the permutation identified

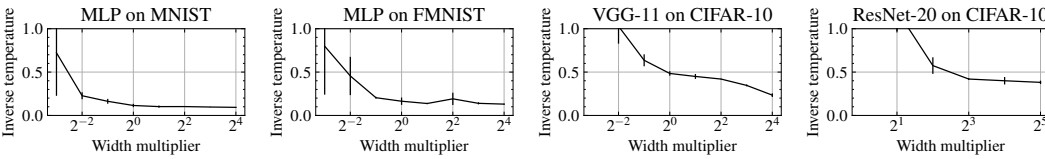

Figure 19: Inverse temperature used for logit calibration when $\lambda = 1/2$.

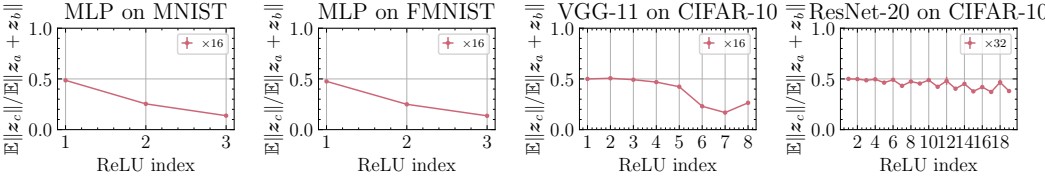

Figure 20: The ratio of activation norms between the merged model and the original models, $\mathbb{E}\|f_\ell(\boldsymbol{x};(\boldsymbol{\theta}_a + \boldsymbol{\theta}_b)/2)\|/\mathbb{E}\|f_\ell(\boldsymbol{x};\boldsymbol{\theta}_a) + f_\ell(\boldsymbol{x};\boldsymbol{\theta}_b)\|$, for $\lambda = 1/2$. The test data are fed into the models.

### F.5 INVERSE TEMPERATURE

We present the inverse temperature values used for calibration when $\lambda = 1/2$ in Figure 19. This inverse temperature corresponds to the factor by which the logits are divided during calibration. As the results indicate, under sufficiently wide conditions, the inverse temperature takes small values, compensating for the reduction in the logit norms caused by LEWC.

### F.6 EXPONENTIAL DECAY OF ACTIVATION NORMS

In Definition 4.1, we showed that the norm of the merged model's output decreases exponentially. In this subsection, we experimentally verify whether this phenomenon actually occurs. The results are presented in Figure 20. For the MLP, as the depth increases, the activation norms of the merged model decrease exponentially, which is consistent with the prediction from LEWC. In contrast, for VGG-11 and ResNet-20, the norms do not decrease exponentially. This is presumably because batch normalization compensates for the variance.

### F.7 SECOND MOMENTS OF RELU INPUTS FOR ALL LAYERS

In this subsection, we present histograms of the square root of the second (uncentered) moment of the ReLU inputs for all layers of each model. The results correspond to an MLP with width scaled by $16\times$, a VGG-11 scaled by $16\times$, and a ResNet-20 scaled by $32\times$. The histograms are shown in Figure 21. The vertical axis is plotted on a logarithmic scale. In addition, the values shown next to "Small input std." indicate the proportion of input dimensions that fall into the leftmost bin. From Figure 21, it can be observed that in every layer, the vast majority of dimensions have a very small second moment around zero.

### F.8 OVERLAP RATIO OF ACTIVE NEURONS WHEN USING PERMUTATIONS

Figure 22 shows the overlap ratio of active neurons in the second ReLU layer when applying the permutations identified by WM. This figure corresponds to Figure 6, where permutations are not used. When permutations are used, the non overlap ratio of active neurons clearly decreases, indicating that most neurons with large output second moments overlap between the two models. We attribute this to the fact that WM searches for permutations that reduce the $L^2$ distance between the weight matrices of the two models, which in turn tends to align their activation patterns.

## F.9 Dependence on Weight Decay

Figure 23 shows the barrier values for permutation-free merging while varying the weight decay coefficient from 0 to 0.003. For sufficiently wide models, smaller weight decay results in smaller barriers.

## F.10 Second Order Optimization

In the main experiments, we used first order optimizers such as SGD and Adam. Here, we additionally examine the case where a second order optimization method is employed. Specifically, we adopt Sophia as a second order optimizer (Liu et al., 2024). The barrier results are shown in Figure 24. For reference, we also report the test accuracy and calibrated loss of the trained models in Figure 25. The training conditions, including hyperparameters, are kept the same as in the first order optimization setting. Therefore, the weight decay is fixed to 0.003.

From Figure 25, we observe that using a second order optimizer improves both accuracy and loss. However, the barriers are smaller when using first order optimizers. With second order optimization, the barriers remain clearly larger than zero for all models except for the MLP trained on MNIST. Nevertheless, the barrier decreases as the width increases, suggesting that LMC may hold when the width is sufficiently large. Since we did not retune hyperparameters for Sophia, it is also possible that using a larger weight decay or a different learning rate would make LMC more likely to emerge under second order optimization.

## F.11 Effect of Skip Connections and Batch Normalization

To clarify the effects of skip connections and batch normalization, we prepared two variants of the MLP: one with skip connections and one with batch normalization layers. For these models, we examined the barrier values and how easily reciprocal orthogonality is satisfied. Skip connections were added by summing each intermediate layer input with its output, for all hidden layers except the input and final layers. The resulting barrier values are shown in Figure 26, where the loss barrier is computed using calibrated loss. As shown in the figure, adding skip connections or batch normalization leads to smaller barriers.

To assess reciprocal orthogonality, we also evaluated the mean norm ratio $\mathbb{E}\|\boldsymbol{W}_\ell^{(a)}\boldsymbol{z}_\ell^{(b)}\|/\mathbb{E}\|\boldsymbol{W}_\ell^{(a)}\boldsymbol{z}_\ell^{(a)}\|$ for each layer, as reported in Figure 27. The results indicate that the presence or absence of skip connections or batch normalization does not substantially change this ratio.

## Use of Large Language Models (LLMs)

In preparing this manuscript, we made use of a large language model (ChatGPT-5, developed by OpenAI). The model was employed exclusively to check the fluency and naturalness of English sentences drafted by the authors. All technical content, mathematical derivations, experimental design, and scientific claims were created and verified by the authors. The authors carefully reviewed and edited all model outputs to ensure accuracy and clarity.

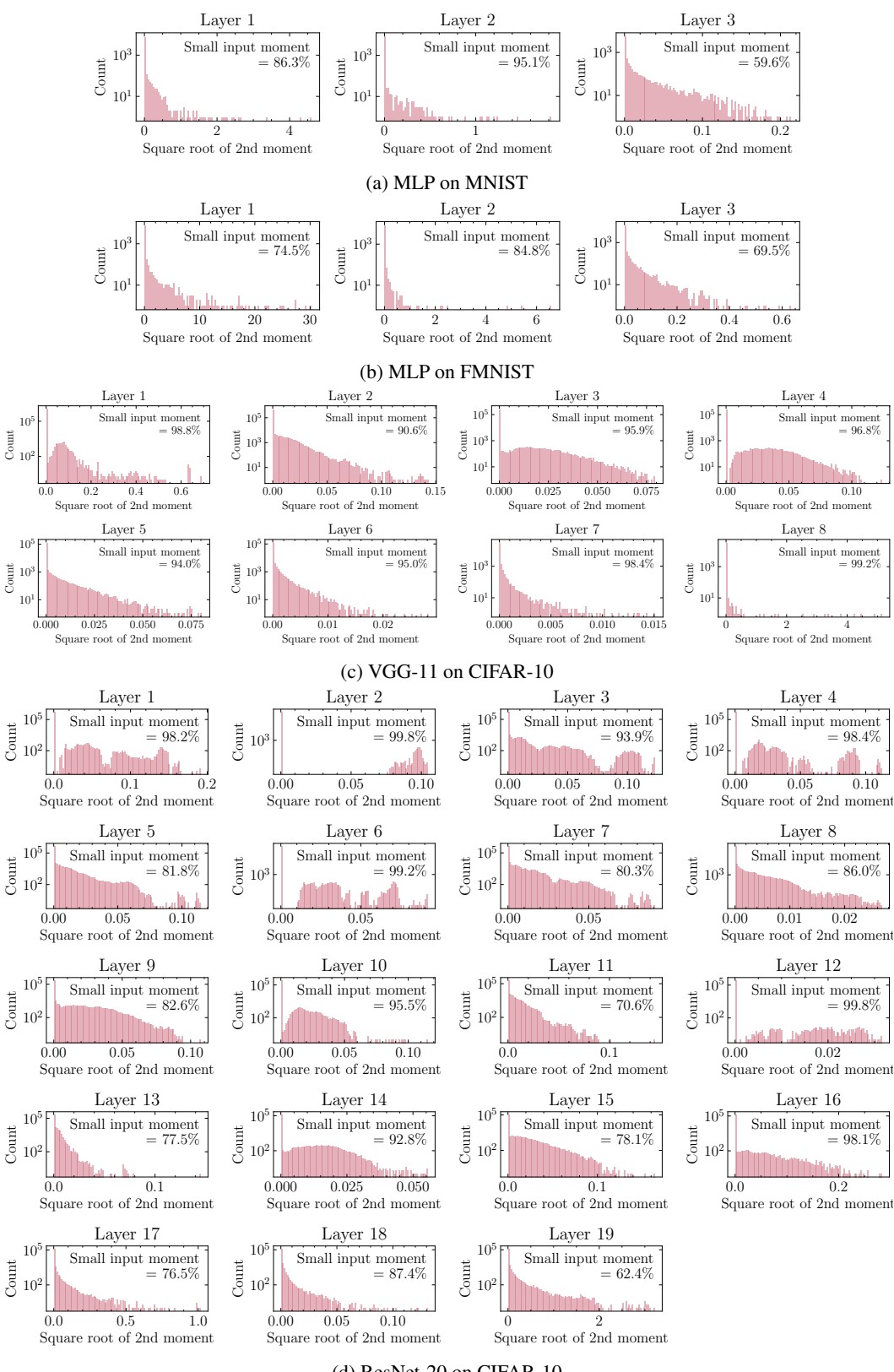

Figure 21: Histogram of the square root of the second (uncentered) moment for each dimension of the ReLU inputs across all layers.

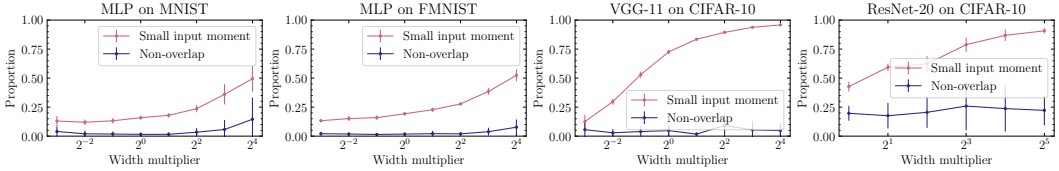

Figure 22: Proportion of input dimensions with a small square root of the second moment ("Small input moment") and the proportion of non-overlapping large moment dimensions ("Non overlap") between two trained models for the second ReLU layer, when using permutations found by WM.

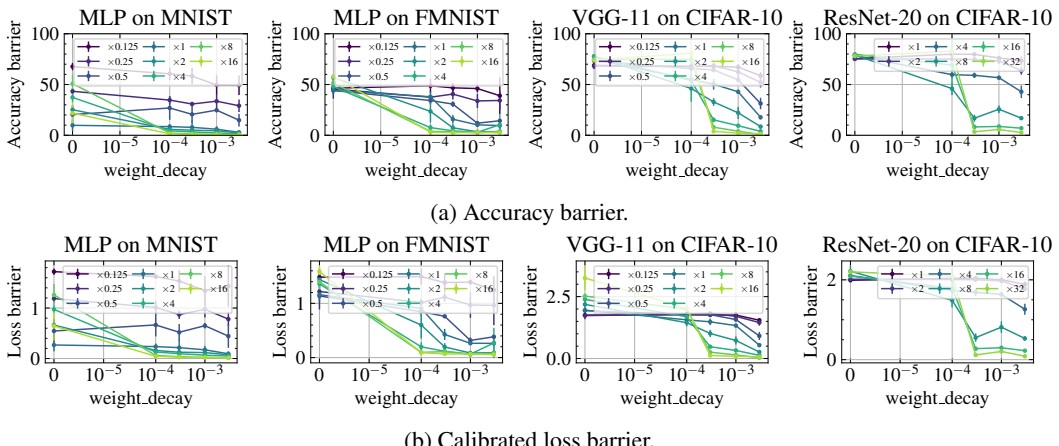

(a) Accuracy barrier.

(b) Calibrated loss barrier.

Figure 23: Accuracy and loss barriers when varying the strength of weight decay.

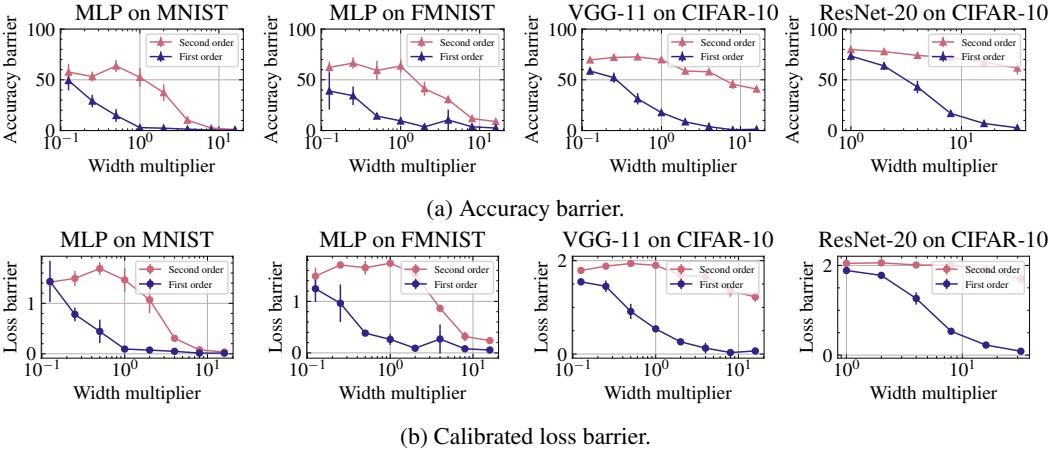

(a) Accuracy barrier.

(b) Calibrated loss barrier.

Figure 24: Accuracy and loss barriers with and without second-order optimization ($\lambda = 1/2$).

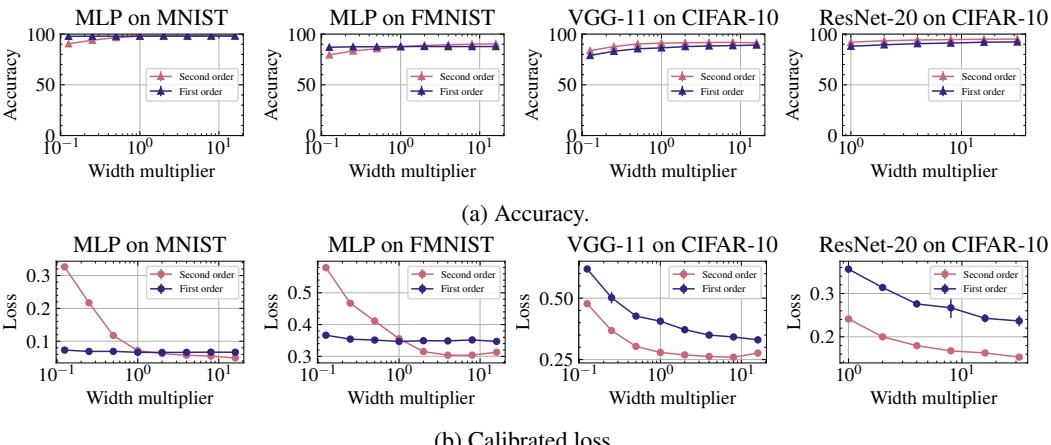

Figure 25: Accuracy and loss of trained models with and without second order optimization.

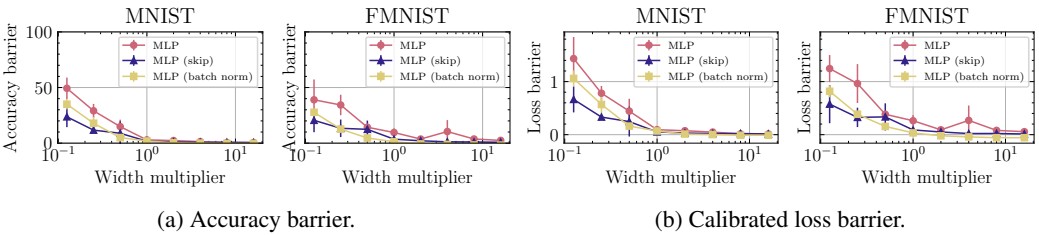

Figure 26: Accuracy and calibrated loss barriers of models trained with and without skip connections or batch normalization.

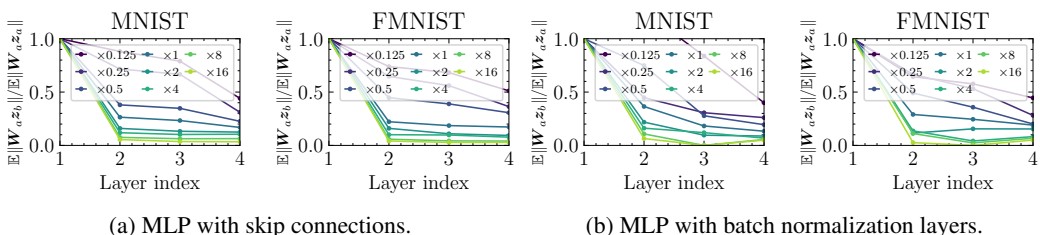

Figure 27: Ratio of mean norms $\mathbb{E}\|\boldsymbol{W}_\ell^{(a)}\boldsymbol{z}_\ell^{(b)}\|/\mathbb{E}\|\boldsymbol{W}_\ell^{(a)}\boldsymbol{z}_\ell^{(a)}\|$ for each layer, for MLPs with skip connections or batch normalization.

