# OpenReview forum: "Do We Really Need Permutations? Impact of Model Width on Linear Mode Connectivity"
_ICLR.cc/2026/Conference — ICLR 2026 Poster_

### Official Review · Reviewer_RKhX · 2025-10-26

**Soundness:** 3
**Presentation:** 3
**Contribution:** 2
**Rating:** 4
**Confidence:** 4

**Summary:**

The paper tackles the task of merging different checkpoints of the same architecture trained from scratch with different seeds. In particular, the paper investigates why linear mode connectivity increases just by increasing the width of the model, without the need for permutations. To explain this, they propose Layerwise Exponentially Weighted Connectivity (LEWC): the intermediate output of the merged model can be expressed as a weighted sum of those of the endpoint models, with the interpolation coefficients decaying exponentially with the widths. The paper derives two sufficient conditions for this phenomenon to arise: ReLU weak additivity and reciprocal orthogonality. The experiments then empirically confirm these ones to hold in several controlled settings, including MLPs, VGGs and ResNets over CIFAR10, MNIST and FMNIST.

**Strengths:**

- The paper is clearly written and easy to read. The exposition is well-structured, and the theoretical derivations are both intuitive and sound.
- The problem is important, both in providing further understanding in loss landscape connectivity and carrying strong implications for model merging, editing and alignment.
- The proposed theoretical framework is novel and interesting. Layerwise Exponentially Weighted Connectivity (LEWC) offers a way to generalize the “averaging as ensembling” point of view that is true for linear model to non-linear deep architectures. The sufficient conditions for LEWC to arise are formally rigorous and make intuitive sense.

**Weaknesses:**

- I have some reservations about the main claim of the paper. The observed permutation-free linear connectivity may stem from overparameterization-induced robustness rather than genuine connectivity. In very wide networks and easy tasks (e.g. CIFAR-10, MNIST), independent checkpoints can lie in flat regions of the loss landscape where large parameter perturbations (like naive weight averaging) don’t affect accuracy. In this perspective, the reported effect might be explained by functional redundancy and flatness rather than the what is stated in the paper. It would be interesting to see the following experiments to better understand if this is the case:
    - Train seeds with constraints that violate the sufficient conditions for LEWC by setting  zero weight decay and spectral regularizers against low-rank. Measure ReLU weak additivity, reciprocal orthogonality and merge success. If merging still works despite not exhibiting relu weak additivity and reciprocal orthogonality then it doesn’t have to do with LEWC and it’s rather just functional redundancy.
    - Compute the norm of the difference between A and B, then traverse a random direction *u* by that same distance and plot the loss barrier throughout. The loss should clearly increase in that direction, but if it’s still low at $\lambda$ = 0.5 then we might be again observing functional redundancy.
- Limited practical utility. The effect appears only for very wide models trained on simple datasets, under settings that are not those used in realistic large-scale systems. Moreover, the need for temperature calibration, strong weight decay, and substantial width multipliers means that the reported phenomenon does not affect practical model-merging pipelines.
- Experimental evidence is limited to toy models and datasets. While this is somewhat common in prior literature, its applicability and theoretical impact would greatly benefit from considering more complex architectures such as ViTs. This could be done by following e.g. [1].

Overall, I find the analysis and insights interesting and worth sharing, but requiring the proposed additional experiments to ensure we are not just observing spurious explanations for unknown, possibly trivial phenomena.

[1] Theus, Alexander, et al. "Generalized Linear Mode Connectivity for Transformers." NeurIPS 2025.

**Questions:**

- To what extent are the observed results a consequence of extreme overparameterization and the smoothness of the loss landscape, rather than a new structural mechanism such as LEWC?
- Do suggested experiments in Weakness 1 confirm LEWC? If these cannot be performed in the rebuttal time frame, what is the expected outcome?
- Modern architectures have moved beyond ReLU. How does the treatment differ if one employs one more modern variant (e.g. SwiGLU)?
- What are the practical implications? requiring strong weight decay and large width multipliers, does this actually inform or help any practical merging or federated learning pipeline?

---

> ### Author Response · Authors · 2025-11-26
>
> We appreciate your thoughtful comments and the time you took to evaluate our work.
>
> - I have some reservations about the main claim of the paper. The observed permutation-free linear connectivity may stem from overparameterization-induced robustness rather than genuine connectivity. In very wide networks and easy tasks (e.g. CIFAR-10, MNIST), independent checkpoints can lie in flat regions of the loss landscape where large parameter perturbations (like naive weight averaging) don't affect accuracy. In this perspective, the reported effect might be explained by functional redundancy and flatness rather than the what is stated in the paper. It would be interesting to see the following experiments to better understand if this is the case:
>
> Thank you for the comment. To clarify whether our results can be explained purely by flatness, we conducted additional experiments. Please see our responses to the two questions below.
>
> - Train seeds with constraints that violate the sufficient conditions for LEWC by setting zero weight decay and spectral regularizers against low-rank. Measure ReLU weak additivity, reciprocal orthogonality and merge success. If merging still works despite not exhibiting relu weak additivity and reciprocal orthogonality then it doesn't have to do with LEWC and it's rather just functional redundancy.
>
> First, LMC does not hold once weight decay is set to zero. This result corresponds to the leftmost setting in the newly added Figure 23. Nevertheless, to follow your suggestion more closely, we also examined the case of zero weight decay combined with an explicit spectral regularizer. Concretely, for VGG-11 and ResNet-20, we replaced batch normalization with spectral normalization and kept all other training conditions unchanged.
>
> Under this setting, VGG-11 failed to train properly, with test accuracy dropping to 11.95 percent. We believe this is due to optimization difficulties such as vanishing gradients. In contrast, ResNet-20 trained successfully, reaching 92.17 percent test accuracy and a calibrated test loss of about 0.24, likely because skip connections stabilize training. Since VGG-11 did not train, we performed merging only for ResNet-20. The merged model achieved 16.49 percent test accuracy and a calibrated test loss of 2.24, indicating that LMC completely fails in this regime. We believe this failure stems from the loss of low rank structure once weight decay is removed, which in turn prevents LEWC and LMC from holding.
>
> - Compute the norm of the difference between A and B, then traverse a random direction u by that same distance and plot the loss barrier throughout. The loss should clearly increase in that direction, but if it's still low at $\\lambda = 0.5$ then we might be again observing functional redundancy.
>
> Thank you for the suggestion. We performed the proposed experiment and added the analysis in Appendix D. While we refer to the manuscript for full details, the main conclusion is as follows. When the width is sufficiently large, adding a random Gaussian direction to the weights changes neither the loss nor the accuracy noticeably. Note that the loss here is uncalibrated. Therefore, the flatness of the raw loss landscape clearly improves as width increases.
>
> However, this improvement in flatness probably does not explain why LMC emerges in our setting, for two reasons.
>
> First, the relationship between two independently trained models $\\boldsymbol{\\theta}\_a$ and $\\boldsymbol{\\theta}\_b$ is not well captured by comparing $\\boldsymbol{\\theta}\_a$ to a randomly perturbed model $\\boldsymbol{\\theta}\_b' := \\boldsymbol{\\theta}\_a + \\|\\boldsymbol{\\theta}\_a - \\boldsymbol{\\theta}\_b\\| \\boldsymbol{u} / \\|\\boldsymbol{u}\\|$, where $\\boldsymbol{u}$ is a random vector sampled from a Gaussian distribution. In particular, the cosine similarity of hidden layer activations is very small between $\\boldsymbol{\\theta}\_a$ and $\\boldsymbol{\\theta}\_b$, whereas it is very high between $\\boldsymbol{\\theta}\_a$ and $\\boldsymbol{\\theta}\_b'$.

---

> > ### Author Response · Authors · 2025-11-26
> >
> > Intuitively, this gap arises because the random perturbation distributes its effect very differently from an independently trained solution. With weight decay, the model distance $d=\\|\\boldsymbol{\\theta}\_a-\\boldsymbol{\\theta}\_b\\|$ does not grow proportionally with width, so the perturbation magnitude remains of comparable scale even in wide networks. When width increases, however, a fixed-norm Gaussian perturbation is spread over many more parameters, so each coordinate of $\\boldsymbol{\\theta}\_a$ receives only a very small addition. In each layer, pre-activations are computed as inner products between an input vector and the rows of the weight matrix. These inner products involve sums over a width-proportional number of terms. The corresponding contribution from the Gaussian noise is also a sum of many independent zero-mean terms, so in sufficiently wide layers its effect concentrates near zero. As a result, the intermediate representations of $\\boldsymbol{\\theta}\_a$ and the randomly perturbed model $\\boldsymbol{\\theta}\_b'$ remain very similar, leading to high cosine similarity.
> >
> > By contrast, the independently trained model $\\boldsymbol{\\theta}\_b$ differs from $\\boldsymbol{\\theta}\_a$ in a structured way rather than as isotropic noise. Its changes are concentrated in specific directions that reflect learned features, so $\\boldsymbol{\\theta}\_b$ can produce substantially different intermediate activations from $\\boldsymbol{\\theta}\_a$. A random perturbation therefore fails to capture the functional relationship between two SGD-trained solutions, which is why flatness measured via random directions is not a reliable explanation for our permutation-free LMC results.
> >
> > Second, if LMC could be explained purely by increased flatness, then we would not observe the exponential decay of intermediate activation norms predicted by LEWC, nor would inverse temperature calibration be necessary for the merged model to match the original test loss. These two points support that the emergence of LMC with increasing width cannot be attributed solely to a generic improvement in flatness.
> >
> > - Limited practical utility. The effect appears only for very wide models trained on simple datasets, under settings that are not those used in realistic large-scale systems. Moreover, the need for temperature calibration, strong weight decay, and substantial width multipliers means that the reported phenomenon does not affect practical model-merging pipelines.
> > - What are the practical implications? requiring strong weight decay and large width multipliers, does this actually inform or help any practical merging or federated learning pipeline?
> >
> > Thank you for raising these concerns. The goal of this paper is not to propose a practical merging algorithm or a federated learning pipeline. Rather, our aim is to clarify the phenomenon that LMC can emerge solely by increasing model width, and to identify the underlying mechanisms behind it. In particular, our analysis shows that this emergence is explained by two properties, weak additivity for ReLU activations and reciprocal orthogonality.
> >
> > At the same time, this perspective suggests potential practical directions. If training procedures or permutation search methods are designed to encourage these two properties, it may become possible to develop new merging or federated learning approaches for independently trained models. In addition, while LEWC predicts that intermediate activation norms may decay exponentially after merging, this issue can be mitigated by appropriately rescaling the merged model’s weights. We view these possibilities as natural follow ups to the present work. We have added a brief discussion of this point in the conclusion to better clarify the intended implications.
> >
> > - Experimental evidence is limited to toy models and datasets. While this is somewhat common in prior literature, its applicability and theoretical impact would greatly benefit from considering more complex architectures such as ViTs. This could be done by following e.g. [1].
> >
> > Thank you for the suggestion. At this stage, we are not able to provide ViT results. The approach in [1] does not come with released code, and we have not yet completed an implementation and evaluation on ViTs. Extending our study to ViTs and other attention based architectures is an important direction for future work.
> >
> > We also note a potential difficulty specific to ViTs. Attention relies on softmax, and the exponential decay of activation or logit norms predicted by LEWC could negatively affect the softmax dynamics. For this reason, we suspect that in architectures with attention, simply merging independently trained models without permutations may be less likely to yield LMC, unless additional mechanisms such as normalization or appropriate rescaling counteract the norm decay.

---

> > > ### Author Response · Authors · 2025-11-26
> > >
> > > - Modern architectures have moved beyond ReLU. How does the treatment differ if one employs one more modern variant (e.g. SwiGLU)?
> > >
> > > Thank you for the comment. We unfortunately did not run experiments with SwiGLU or other modern activations, so what follows is a conjecture. We expect that LEWC could still emerge as width increases, provided that an analogue of weak additivity holds for the chosen activation. In the paper, we explained weak additivity for ReLU through the observation that widening reduces the overlap ratio of active neurons. With sufficiently strong weight decay, a similar sparsification or effective linearization effect may occur beyond ReLU, so approximate linearity may also hold for activations such as SwiGLU.
> > >
> > > A potential caveat is that SwiGLU includes a smooth gating function like Swish, so the exponential norm reduction predicted by LEWC could have a stronger negative effect than in the ReLU case. However, if normalization layers such as batch normalization or layer normalization are present, they would compensate for norm changes, and the issue might be mitigated.

---

### Official Review · Reviewer_3r8c · 2025-10-31

**Soundness:** 3
**Presentation:** 3
**Contribution:** 2
**Rating:** 4
**Confidence:** 4

**Summary:**

This paper argues that sufficiently wide neural networks achieve linear mode connectivity without permutation alignment. While permutation-based methods require networks to have a minimum width, the authors claim that for a wide enough network the permutation search itself becomes unnecessary and provide insights on why it emerges. The authors extend layerwise linear feature connectivity (LLFC) by introducing layerwise exponentially weighted connectivity (LEWC), positing that the intermediate activations can be expressed as an exponentially weighted average of those of the individual models. They explain the emergence of LEWC with weak additivity of ReLU activations (same as LLFC) and reciprocal orthogonality.

**Strengths:**

- The exponential derivation and the LEWC framework represent a meaningful extension to the layerwise linear feature connectivity, novel to the best of my knowledge.
- The paper contributes to emerging literature questioning permutation alignment as necessary for mode connectivity, challenging a mainstream assumption in the field.
- The paper provides a principled explanation for why LMC is more easily achieved in wider networks through the lens of reciprocal orthogonality, moving beyond the intuition that "more width provides more permutations to search." This offers a different perspective on the role of overparameterization in mode connectivity.
- The empirical framework and justification is sound.

**Weaknesses:**

-The paper makes it sound like the networks are widened post-hoc after training. The title and framing suggest a novel widening procedure (potentially post-hoc expansion of trained models, which would be very exciting), but the paper appears to simply experiment with networks that are already wide from initialization, this should be made clear. I believe post-hoc widening could be a very interesting addition to impose reciprocal orthogonality and low-rank structure.
- The experiments inherit the limited scope of prior work and need expansion, some suggestions are included in the questions
- I think the paper could benefit from an expanded discussion in relation to over-parameterized networks (e.g. Simsek et al, 2021).

**Questions:**

- Could you provide more analysis of the temperature parameter in Section 3.1? Is there a relationship between optimal temperature and network width/depth/alpha? How sensitive are the results to this choice?
- I find the low-rank argument insightful. Given LMC is sensitive to optimization hyper-parameters (Altıntaş, et al. 2024) I would be interested to see a comparison between first and second-order methods in this line. Another experiment could be using no weight decay, since it is also a common practice in training Cifar-scale Resnets.
- Have the authors investigated how the permutation-based methods affect the overlapping dimensions? Though not critical, this could be a good insight of the paper. I think an analysis in this line could support the claims in the paper.
- I would be interested to see if the connectivity breaks gradually over layers and hence if there is a relationship between the width expansion factor and layer depth?
- I think it would be useful if the authors commented on how normalization layers and residual connections interfere with the reciprocal orthogonality. The authors could experiment with this in the MLP architecture.
- Even though the authors elaborate on LLFC (and commutativity not holding in their seeting) in the appendix I believe the paper could benefit from a broader discussion of the implications of LLFC vs LEWC as well as the emergence of LEWC in the spawning setting (Frankle et al, 2018). This could be an ablation of the paper.

I find this line of research and the authors' contribution quite interesting and I am open to increasing my score if the authors can situate the implications of their work better within the existing literature.

---

> ### Author Response · Authors · 2025-11-26
>
> Thank you for your careful review and constructive feedback.
>
> - The paper makes it sound like the networks are widened post-hoc after training. The title and framing suggest a novel widening procedure (potentially post-hoc expansion of trained models, which would be very exciting), but the paper appears to simply experiment with networks that are already wide from initialization, this should be made clear. I believe post-hoc widening could be a very interesting addition to impose reciprocal orthogonality and low-rank structure.
>
> We apologize for the potentially misleading title. We have changed it to “Do We Really Need Permutations? Impact of Model Width on Linear Mode Connectivity.” As you pointed out, post hoc widening would be an interesting research direction. However, the aim of this work is to analyze how the model width during training affects LMC, so we believe that post hoc widening is better pursued as a follow up to this study.
>
> - I think the paper could benefit from an expanded discussion in relation to over-parameterized networks (e.g. Simsek et al, 2021).
>
> - I find this line of research and the authors' contribution quite interesting and I am open to increasing my score if the authors can situate the implications of their work better within the existing literature.
>
> Thank you for the comment. We believe our results can be positioned within the literature on overparameterized networks as follows.
>
> A broad trend in prior work is that increasing width tends to simplify the loss landscape of deep neural networks and makes the set of optimal solutions more connected. Nguyen (2021) and Simsek et al. (2021) pointed out that the topology of the optimal solution set can change drastically by adding only a neuron. Specifically, Nguyen (2021) proved that for deep networks, having a single wide layer of width $n+1$, where $n$ is the number of training samples, is sufficient to ensure that all sublevel sets of the training loss are connected. Simsek et al. (2021) further showed that adding one extra neuron per layer is enough to turn discrete minima into a single connected manifold. Liang et al. (2018) showed that adding one special neuron to each layer can eliminate all bad local minima, so that every local minimum becomes a global minimum. Subsequent works have also reported phase transition like behavior as width increases. For example, Li et al. (2022) showed that there exists a critical width $m^\ast$ such that when $m \ge m^\ast$, all suboptimal basins disappear and only optimal basins remain. More recently, Kim et al. (2025) established that in regularized two-layer networks, once the width exceeds a certain threshold, the optimal solution set becomes connected in parameter space.
>
>
> These results indicate that widening changes the structure of the optimal set, and in particular for regularized models it eventually becomes connected, which implies mode connectivity. Our work goes one step further by empirically showing that, after appropriately calibrating the loss via inverse temperature scaling, the stronger property of linear mode connectivity holds in sufficiently wide networks. Existing theoretical studies do not explicitly consider such inverse temperature calibration, but this transformation does not alter accuracy and can therefore be regarded as permissible when discussing solution connectivity with respect to the final target metric. From this perspective, our contribution is to highlight a natural extension of prior landscape geometry results, namely to ask whether widening yields linear, not merely continuous, connectivity under loss transformations that preserve accuracy. We believe this viewpoint is useful for understanding training dynamics and complements the existing overparameterization literature.
>
> We have added the above discussion in Appendix A.
>
> 1. Nguyen, Quynh. "A note on connectivity of sublevel sets in deep learning." arXiv preprint arXiv:2101.08576 (2021).
> 2. Simsek, Berfin, et al. "Geometry of the loss landscape in overparameterized neural networks: Symmetries and invariances." International Conference on Machine Learning. PMLR, 2021.
> 3. Liang, Shiyu, et al. "Adding one neuron can eliminate all bad local minima." Advances in Neural Information Processing Systems 31 (2018).
> 4. Li, Dawei, Tian Ding, and Ruoyu Sun. "On the benefit of width for neural networks: Disappearance of basins." SIAM Journal on Optimization 32.3 (2022): 1728-1758.
> 5. Kim, Sungyoon, Aaron Mishkin, and Mert Pilanci. "Exploring The Loss Landscape Of Regularized Neural Networks Via Convex Duality." The Thirteenth International Conference on Learning Representations.

---

> > ### Author Response · Authors · 2025-11-26
> >
> > - The experiments inherit the limited scope of prior work and need expansion, some suggestions are included in the questions
> >
> > Thank you for the comment. Based on your suggestions, we conducted additional experiments. Please see the details below.
> >
> > - Could you provide more analysis of the temperature parameter in Section 3.1? Is there a relationship between optimal temperature and network width/depth/alpha? How sensitive are the results to this choice?
> >
> > Thank you for the comment. To analyze this point, we added Figure 19, which reports the inverse temperature values actually used for calibration, and Figure 20, which shows the layerwise decay ratio of intermediate activation norms, namely $\\mathbb{E}\\|f\_\\ell(\\boldsymbol{x}; (\\boldsymbol{\\theta}\_a + \\boldsymbol{\\theta}\_b)/2)\\| \\,/\\, \\mathbb{E}\\|f\_\\ell(\\boldsymbol{x}; \\boldsymbol{\\theta}\_a) + f\_\\ell(\\boldsymbol{x}; \\boldsymbol{\\theta}\_b)\\|$. Both figures correspond to the case $\\lambda = 1/2$, and Figure 20 shows only the widest setting. Therefore, the rightmost result in Figure 19 (the widest model) should be compared to the last value in Figure 20 (the final layer). These two quantities are nearly the same, indicating that the inverse temperature is chosen to compensate for the norm decay at the output layer. This suggests that the temperature parameter is largely determined by the decay factor at the final layer. We also observe that in MLPs the norm decays monotonically toward the output layer, whereas in VGG 11 and ResNet 20 the decay is much weaker because batch normalization corrects the norms.
> >
> > In LEWC, the decay of intermediate layer norms depends only on depth and not on width. Hence, the temperature parameter should become larger for deeper networks. However, when batch normalization is present, norms are partially corrected, so the decay does not necessarily grow exponentially with depth. In addition, the interpolation coefficient $\\lambda$ (we assume the $\\alpha$ in the comment refers to $\\lambda$) affects the decay rate. When $\\lambda$ is closer to 0 or 1, the decay across depth becomes milder, so the required temperature should be smaller than in the case $\\lambda = 1/2$.
> >
> >
> > - I find the low-rank argument insightful. Given LMC is sensitive to optimization hyper-parameters (Altıntaş, et al. 2024) I would be interested to see a comparison between first and second-order methods in this line. Another experiment could be using no weight decay, since it is also a common practice in training Cifar-scale Resnets.
> >
> > Thank you for the comment. While this paper mainly focuses on first-order optimization methods such as SGD, we added a comparison with a second-order method to clarify the behavior under different training conditions. Specifically, we included new results in Appendix F.10 for models trained with Sophia as a second-order optimizer (Liu et al., 2024). We used the same hyperparameters as in the first-order setting. The results show that LMC becomes harder to satisfy under second-order optimization. However, the barrier still decreases as the width increases, suggesting that LMC may hold for sufficiently wide models. In addition, since we did not tune hyperparameters for Sophia, LMC might become more likely with appropriate tuning, for example by using a stronger weight decay. Investigating when LMC holds under second-order optimization is an important direction for future work.
> >
> > We also added a new plot in Figure 23 that reports the barriers when weight decay is removed. The leftmost setting in the figure corresponds to zero weight decay. As predicted in our analysis, the barriers are large in this case, indicating that LMC does not hold without weight decay.
> >
> > - Have the authors investigated how the permutation-based methods affect the overlapping dimensions? Though not critical, this could be a good insight of the paper. I think an analysis in this line could support the claims in the paper.
> >
> > Thank you for the comment. We have added an analysis in Appendix F.8 that reports the overlap ratio when using the permutations identified by WM. As shown in the figure, applying permutations substantially increases the overlap ratio, which is clearly different from the case without permutations. We believe this trend is consistent with the difference between commutativity in LLFC and reciprocal orthogonality in LEWC. The former requires the two weight matrices to be close, whereas the latter requires them to be different, namely approximately orthogonal. A high overlap ratio of neurons with large output second moments indicates that the weight matrices are closer to each other, meaning that the two models become more functionally aligned.

---

> > > ### Author Response · Authors · 2025-11-26
> > >
> > > - I would be interested to see if the connectivity breaks gradually over layers and hence if there is a relationship between the width expansion factor and layer depth?
> > >
> > > Weak additivity for ReLU activations and reciprocal orthogonality in LEWC are also properties that hold only approximately, rather than exactly. Therefore, we expect the connectivity to break more as the depth increases. On the other hand, widening the model should make these properties easier to satisfy, so we conjecture that deeper models require larger widths for LEWC, and hence LMC, to hold.
> > >
> > > - I think it would be useful if the authors commented on how normalization layers and residual connections interfere with the reciprocal orthogonality. The authors could experiment with this in the MLP architecture.
> > >
> > > Thank you for the suggestion. We have added experimental results in Appendix F.11 that examine the effects of adding skip connections and batch normalization layers to the MLP. Compared to the original MLP, we find that the barriers are smaller when skip connections or batch normalization are present. We also observe that reciprocal orthogonality holds regardless of whether skip connections or batch normalization are used. In particular, we do not see evidence that these components interfere with reciprocal orthogonality.
> > >
> > > - Even though the authors elaborate on LLFC (and commutativity not holding in their setting) in the appendix I believe the paper could benefit from a broader discussion of the implications of LLFC vs LEWC as well as the emergence of LEWC in the spawning setting (Frankle et al., 2018). This could be an ablation of the paper.
> > >
> > > Thank you for the comment. You are right that we did not include a qualitative discussion of the relationship between LLFC and LEWC in the main text. We have added this discussion in Appendix C. The specific differences are summarized in our response to the overall comments at the beginning of the rebuttal, so please refer to that section as well.

---

### Official Review · Reviewer_EvXf · 2025-11-01

**Soundness:** 4
**Presentation:** 3
**Contribution:** 3
**Rating:** 6
**Confidence:** 4

**Summary:**

This paper explores the narrative around permutation symmetries and linear mode connectivity (LMC) from a new angle, finding that permutation symmetries are not necessary to achieve LMC for very wide models if one is allowed to recalibrate the temperature of the softmax (a much smaller modification). The paper then introduces layerwise exponentially weighted connectivity (LEWC), a nonlinear form of connectivity which is satisfied when the activations of the model decay exponentially as the layer index increases. The key observation is that low rank weights reduce the conflict between merged neurons and (relatively) large settings of weight decay promote low-rank weights.

**Strengths:**

1. The key observation of the paper is very scientifically interesting and portrays linear mode connectivity and permutation symmetries in a new light.
2. The paper also contains theoretical analysis justifying why LEWC can be expected to emerge for very wide models.
   1. The theoretical analysis is based on two properties:
   2. Weak Additivity for ReLU Activations (roughly, interpolating the inputs of ReLUs is the same as interpolating their outputs)
   3. Reciprocal Orthogonality (roughly, each model's features live in the "null space" of the other)
3. The above properties are tested in practice and found to emerge as the width increases. Both properties get strong as the width increases!

**Weaknesses:**

1. The main experiments use a slightly stronger than normal choice of weight decay (3e-3). I believe the typical values are roughly 1e-4 MNIST and 5e-4 for CIFAR, (although I am not familiar with the history of these choices). This is not a major weakness, but it does complicate the narrative since permutation-based methods still work when the weight decay is not so high.

**Questions:**

1. Do permutation methods like Git Re-Basin continue to work with typical choices of weight decay? If so, is the mechanism by which Git Re-Basin "fixes" the calibration of the model? Why doesn't LEWC continue to appear between the permuted models?
2. Do you think the re-calibration step could further reduce the barriers for models merged with permutation methods in settings where LEWC does not arise organically (low weight decay)?
3. Can you record the (midpoint) accuracies/losses of the models in numerical form? It it useful to see the exact numbers since differences in accuracy can be quite small while staying meaningful (for example).
4. The larger setting of weight decay is important to establish the low-rank structure which allows the permutation-free merging to succeed. Given the merging does not succeed when the weight decay is set to a lower value of 1e-4, I think it would be interesting to see how the loss/accuracy barriers change as the weight decay decreases.

---

> ### Author Response · Authors · 2025-11-26
>
> Thank you for the insightful suggestions, which helped us improve the manuscript.
>
> - Do permutation methods like Git Re-Basin continue to work with typical choices of weight decay? If so, is the mechanism by which Git Re-Basin "fixes" the calibration of the model? Why doesn't LEWC continue to appear between the permuted models?
>
> Yes. Permutation-based methods remain effective under standard weight decay values. For Git Re-Basin, (i.e., permutation based merging), calibration is not needed.
>
> The objective of permutation search methods such as WM is to align the two models by making their weight matrices close, that is, $\\boldsymbol{W}\_\\ell^{(a)} \\approx \\boldsymbol{W}\_\\ell^{(b)}$ for every layer $\\ell$. As a result, the merged weight matrices stay close to those of the original models, so activation norms do not substantially decay across layers. In contrast, in permutation free merging, the weight matrices are not aligned, so averaging them reduces their norms. This leads to smaller activation norms at each layer, and in deep networks the output norm can decrease exponentially. This is why the loss becomes large without calibration in the permutation-free setting.
>
> We believe LEWC does not appear after applying permutations because of the difference between LLFC and LEWC. LLFC requires the two models to be close in weight space, whereas LEWC requires them to be highly different, namely approximately orthogonal. Since permutations make the models close, LLFC is encouraged while LEWC no longer emerges. We provide a more detailed discussion in our overall response.
>
> - Do you think the re-calibration step could further reduce the barriers for models merged with permutation methods in settings where LEWC does not arise organically (low weight decay)?
>
> We do not expect a noticeable reduction. As explained above, permutation based methods do not induce exponential decay of activations, so recalibration should not change the barrier substantially. Although this is not included in the paper, we experimentally confirmed that recalibration reduces the barrier only marginally when permutations are used.
>
> - Can you record the (midpoint) accuracies/losses of the models in numerical form? It it useful to see the exact numbers since differences in accuracy can be quite small while staying meaningful (for example).
>
> We have added numerical values of the barriers in Table 1. This makes the midpoint accuracies and losses explicit even when the differences are small.
>
> - The larger setting of weight decay is important to establish the low-rank structure which allows the permutation-free merging to succeed. Given the merging does not succeed when the weight decay is set to a lower value of 1e-4, I think it would be interesting to see how the loss/accuracy barriers change as the weight decay decreases.
>
> Thank you for the suggestion. We have added results in Appendix F.9 showing how the loss and accuracy barriers change as weight decay decreases.

---

### Official Review · Reviewer_8AS9 · 2025-11-01

**Soundness:** 3
**Presentation:** 4
**Contribution:** 3
**Rating:** 6
**Confidence:** 4

**Summary:**

In the context of model merging and parameter permutations, this work shows that contrary to prevailing thought, increasing network width  is sufficient in and of itself (without permutation alignment of the parameters) to make networks more linearly connected (more merge-able via weight interpolation).  The authors show that a) increasing width reduces barriers (loss barriers require softmax recalibration), b) activations become linearly connected (remain more similar when linearly interpolated) with depth, dependent on two conditions (weak additivity and reciprocal orthogonality), and c) activations are lower rank with less "overlap" as width increases. Reducing weight decay (which increases rank) counteracts this effect.

**Strengths:**

The work presents a novel and quite interesting finding, and backs it up with a predictive theory along with empirical evidence. The structure of the paper is well thought out and covers many of the gotchas and questions one may have going in - e.g. batch norm statistics are handled properly using REPAIR, and many efforts are made to distinguish reciprocal orthogonality from commutativity in Zhou et al. (2023). Overall there are many good things going on and I would like the authors to push a bit further beyond establishing that width-induced LMC exists.

**Weaknesses:**

I have two issues with this work. One issue is that the theory and experiments are somewhat detached as the theoretical setup and some predictions (depth particularly) don't have solid connections with the empirical evidence. Some of the definitions/conditions (e.g. low rank weights leading to LMC) feel arbitrary and accidental rather than clearly supported by intuition and evidence. The other issue is that I'm not sure what to conclude - can we predict when LMC will arise due to width alone? What are the implications (practical or otherwise) of the findings?

Currently, the empirical results cannot rule out the possibility that there is a lower floor on how small the barrier becomes with increased width. First, for the higher width scales, it is important to report the actual loss or accuracy barriers (such as in a table, zoomed-in plot, or plotting width vs max barrier) since "LMC" in the permutation literature (e.g. Ainsworth et al.) requires "essentially no barrier".

Along this line, I don't find the random permutations experiment (Figure 13) useful - in a sense, two randomly initialized and trained networks are already randomly permuted with respect to each other, since the probability of initializing $\theta$ is equal to that of $\pi(\theta)$ for any permutation $\pi$. Instead, I would like to know whether as width increases, the difference between barriers from a known good permutation and a random one goes to 0. For the known good permutation, one could use permutation-finding algorithms, or networks from the same initialization that are LMC (Frankle et al. 2020).

I find the argument in 5.2 (low rank structure reduces overlap) somewhat unconvincing - first, the intuitive example is coordinate-aligned and I don't see how the same applies to arbitrary low-rank transformations which are not coordinate-aligned. It's possible that activations are coordinate-aligned in practice due to ReLU being a coordinate-aligned transform, which figure 6 supports. In general, I think a statement like "width leads to sparser activations, which induces weak additivity" would be stronger than the current "low rank" explanation.

Small problems with the presentation:
- Missing citations for line 59: "In addition, several studies have shown that widening ResNet-50 trained on the ImageNet dataset improves the test accuracy of the merged model when interpolating the weights of two trained models with permutation"
- The plot colors are hard to read for the different layer widths - maybe consider using a graduated palette (e.g. viridis)?
- The "standard deviation with respect to 0" (defined in figu re 5) should be called the square root of the second (uncentered) moment
- I suggest moving figure 5 to the appendix in favour of figure 9 - section 5.4 (reduced weight decay stops the effect) adds a much-needed ablation in my opinion. Actually, more ablations would really help establish that weak additivity and reciprocal orthogonality are necessary and sufficient conditions for LMC sans permutation.

**Questions:**

An interesting implication predicted by LEWC is that the exponential decay in activation magnitude as depth increases. This point isn't really touched on in the main text, and I would like to know if the exponential decay in the magnitude of activations from intermediate layers occurs as predicted:
- What are the actual temperature values in fig 2 after calibration? Is LEWC predictive of the magnitude of these temperature values (i.e. does increasing depth make the scale of the logits smaller)?
- Similarly, for fig 3 what does a plot of depth versus magnitude of the interpolated vector look like?

It isn't immediately obvious from the main text alone what the distinction is between the definition of reciprocal orthogonality and commutativity from Zhou et al. (2023) - in particular, what is the significance of the differences between the experiments for figure 8 in section 5.3 (empirically verifying reciprocal orthogonality) and figure 10 (empirically verifying commutativity doesn't hold)?

Could the authors make a connection between LMC with increasing width, and the neural tangent kernel at the infinite width limit?

---

> ### Author Response · Authors · 2025-11-26
>
> Thank you for your constructive comments. We would like to answer your questions and concerns as follows.
>
> - One issue is that the theory and experiments are somewhat detached as the theoretical setup and some predictions (depth particularly) don't have solid connections with the empirical evidence.
>
> - An interesting implication predicted by LEWC is that the exponential decay in activation magnitude as depth increases. This point isn't really touched on in the main text, and I would like to know if the exponential decay in the magnitude of activations from intermediate layers occurs as predicted:
>
> - Similarly, for fig 3 what does a plot of depth versus magnitude of the interpolated vector look like?
>
> Thank you for your comment regarding the model depth. To examine whether the activation norms indeed exhibit exponential decay with respect to depth, we have added experimental results in Appendix F.6, where we analyze how the activation norms change as the model depth increases. In the MLP, we observe an exponential decay of the norms, which is consistent with the prediction from LEWC. In contrast, for VGG-11 and ResNet-20, no such exponential decay is observed, presumably because batch normalization compensates for the variance (i.e. norm).
>
> - What are the actual temperature values in fig 2 after calibration? Is LEWC predictive of the magnitude of these temperature values (i.e. does increasing depth make the scale of the logits smaller)?
>
> We have added the inverse temperature values actually used for calibration in Appendix F.5. As shown in the newly included figure, the inverse temperature becomes small when the networks are sufficiently wide, indicating that it compensates for the exponential decrease in logit norms induced by LEWC.
>
>
> - Some of the definitions/conditions (e.g. low rank weights leading to LMC) feel arbitrary and accidental rather than clearly supported by intuition and evidence.
>
> - I find the argument in 5.2 (low rank structure reduces overlap) somewhat unconvincing-first, the intuitive example is coordinate-aligned and I don't see how the same applies to arbitrary low-rank transformations which are not coordinate-aligned. It's possible that activations are coordinate-aligned in practice due to ReLU being a coordinate-aligned transform, which figure 6 supports. In general, I think a statement like "width leads to sparser activations, which induces weak additivity" would be stronger than the current "low rank" explanation.
>
> Thank you for the comment. We agree that explaining the phenomenon in terms of sparsity of activations is a stronger and more direct statement than attributing it solely to low rank weights. Indeed, weak additivity for ReLU activations is grounded more directly in the fact that pre-activations become sparse, rather than in low rankness itself. To clarify this point and avoid over stating the role of low rank structure, we added a footnote in Section 5.4.
>
> - The other issue is that I'm not sure what to conclude - can we predict when LMC will arise due to width alone? What are the implications (practical or otherwise) of the findings?
>
> We apologize for the lack of clarity in our original conclusion. By examining whether the two properties we identified, namely weak additivity for ReLU activations and reciprocal orthogonality, are satisfied as the network width increases, it would be possible to predict whether LMC will hold when the width is scaled up.
>
> Furthermore, the focus of this work is to analyze the phenomenon that LMC emerges simply by increasing the width. Through our analysis, we have clarified which underlying mechanisms give rise to LMC. In future work, by developing permutation-search methods or training procedures that make these properties more likely to hold, it may become possible to establish new merging methods for independently trained models that have not been achievable before. We have added a brief discussion of this point in the conclusion.
>
> - First, for the higher width scales, it is important to report the actual loss or accuracy barriers (such as in a table, zoomed-in plot, or plotting width vs max barrier) since "LMC" in the permutation literature (e.g. Ainsworth et al.) requires "essentially no barrier".
>
> Thank you for the comment. To clarify whether the barriers truly become zero, we have added figures in Appendix F.3 and Table 1 that show the accuracy and loss barriers. As the results indicate, when the width is sufficiently large, the calibrated loss barrier becomes zero even without using permutations.

---

> ### Author Response · Authors · 2025-11-26
>
> - Along this line, I don't find the random permutations experiment (Figure 13) useful - in a sense, two randomly initialized and trained networks are already randomly permuted with respect to each other, since the probability of initializing $\theta$ is equal to that of $\pi(\theta)$ for any permutation $\pi$. Instead, I would like to know whether as width increases, the difference between barriers from a known good permutation and a random one goes to 0. For the known good permutation, one could use permutation-finding algorithms, or networks from the same initialization that are LMC (Frankle et al. 2020).
>
> In response to your comment, we have added results in Appendix F.4 that compare the barriers obtained using the permutation identified by WM and those obtained using random permutations. As expected, we find that the difference between these barriers approaches zero when the model width is sufficiently large.
>
> - Missing citations for line 59: "In addition, several studies have shown that widening ResNet-50 trained on the ImageNet dataset improves the test accuracy of the merged model when interpolating the weights of two trained models with permutation"
>
> We apologize for missing citations. We have added the appropriate citations.
>
> - The plot colors are hard to read for the different layer widths - maybe consider using a graduated palette (e.g. viridis)?
>
> Thank you for your suggestion. We have updated all plots to use the Viridis color palette for improved readability.
>
>
> - The "standard deviation with respect to 0" (defined in figure 5) should be called the square root of the second (uncentered) moment
>
> Thank you for the suggestion. We have replaced the term “standard deviation with respect to 0” with “the square root of the second (uncentered) moment” throughout the manuscript.
>
> - I suggest moving figure 5 to the appendix in favour of figure 9 - section 5.4 (reduced weight decay stops the effect) adds a much-needed ablation in my opinion. Actually, more ablations would really help establish that weak additivity and reciprocal orthogonality are necessary and sufficient conditions for LMC sans permutation.
>
> Thank you for the suggestion. Since the page limit is expanded from 9 pages to 10 pages for the rebuttal and camera-ready versions, we chose not to remove Figure 5 and instead moved the analysis of the weak weight decay setting (originally in the appendix) into Section 5.4 of the main text.
>
>
> - It isn't immediately obvious from the main text alone what the distinction is between the definition of reciprocal orthogonality and commutativity from Zhou et al. (2023) - in particular, what is the significance of the differences between the experiments for figure 8 in section 5.3 (empirically verifying reciprocal orthogonality) and figure 10 (empirically verifying commutativity doesn't hold)?
>
> In short, reciprocal orthogonality requires the two models to have sufficiently different weight matrices, namely that cross terms such as $\\boldsymbol{W}\_\\ell^{(a)}\\boldsymbol{z}\_{\\ell-1}^{(b)}$ are close to zero. By contrast, commutativity is a condition that is favored when the two weight matrices are sufficiently close to each other. Therefore, the two properties are conceptually very different and are not expected to hold simultaneously. The reason we observe the pattern in Figures 8 and 10 in the original manuscript is that, without permutations, the two independently trained models remain far apart in weight space, which supports reciprocal orthogonality but makes commutativity fail. We provide a more detailed explanation in our overall response.
>
>
> - Could the authors make a connection between LMC with increasing width, and the neural tangent kernel at the infinite width limit?
>
> Thank you for the comment. As we understand it, the neural tangent kernel describes the infinite-width limit in which, when training with a sufficiently small learning rate, the parameters of a trained model stay close to their initialization. As a consequence, the model output can be well approximated by a first-order Taylor expansion with respect to the parameters. However, this regime requires the learning rate to be very small so that the trained parameters remain near initialization. In our experiments, we do not use such small learning rates, and it is unlikely that the trained models stay close to their initial parameters. Therefore, we do not think our results directly establish a connection between NTK and LMC. Explaining our observations would likely require a framework different from NTK.

---

> > ### Comment · Reviewer_8AS9 · 2025-11-27
> >
> > Thanks for the detailed reply and the additional results in appendix F. I still feel there is somewhat of a disconnect between the theory and the evidence. For example: reciprocal orthogonality implies exponential decay in the activations (assuming no batch norm), but evidence showing exponential decay does not necessarily imply reciprocal orthogonality (a simpler explanation for decay could be just that linear interpolation reduces norms). The plots in appendix F also show decay, but it isn't clear if the decay is exponential and can be predicted from the theoretical mechanism, or if the decay could be attributed to something else.
> >
> > In this vein, I agree with reviewer RKhX's first point that flatness may be a simpler explanation, and I'm not sure appendix D fully addresses this. While a random Gaussian perturbation won't look like interpolation between two trained networks, one could also do something like step in the opposite direction of interpolation (take $\lambda = 1.5$ or $\lambda = -0.5$) and see if those networks are LMC with the originals.
> >
> > Having said that, I don't see a distinction to be made between "real" and "artificial" linear connectivity, just that the proposed mechanism for why linear connectivity occurs remains up for debate. I still think the observed empirical phenomenon is real and valuable to note. Hence, I will keep my score.

---

> > > ### Author Response · Authors · 2025-11-28
> > >
> > > Thank you for your comments. We understand your concerns as consisting of the following three points:
> > >
> > > 1. The relationship between the decay of activation norms and reciprocal orthogonality/LEWC
> > > 2. Distinguishing our explanation from a simpler explanation based on flatness
> > > 3. The behavior outside the interpolation interval (e.g., $\\lambda = 1.5, -0.5$)
> > >
> > > Below we respond to each point in turn.
> > >
> > > ## Relationship between the decay of activation norms and reciprocal orthogonality/LEWC
> > >
> > > As you pointed out, exponential decay of activation norms by itself does not imply that LEWC holds. Intuitively, reciprocal orthogonality means that the two weights are orthogonal in an appropriate sense, and as long as reciprocal orthogonality holds, the norms will decay layer by layer. Therefore, observing exponential decay is one piece of evidence suggesting that reciprocal orthogonality may hold, but it does not by itself guarantee that reciprocal orthogonality is satisfied.
> > >
> > > However, in our work we do not rely solely on activation norm decay to claim that LEWC and reciprocal orthogonality holds. We verify LEWC and reciprocal orthogonality in several independent ways. For example, in Figure 7 we directly measure how small $\\|\\boldsymbol{W}\_\\ell^{(a)} \\boldsymbol{z}\_{\\ell-1}^{(b)}\\|$ is relative to $\\|\\boldsymbol{W}\_\\ell^{(a)} \\boldsymbol{z}\_{\\ell-1}^{(a)}\\|$.
> > > The results show that, as we increase the width of the network,  $\\|\\boldsymbol{W}\_\\ell^{(a)} \\boldsymbol{z}\_{\\ell-1}^{(b)}\\|$ becomes significantly smaller than $\\|\\boldsymbol{W}\_\\ell^{(a)} \\boldsymbol{z}\_{\\ell-1}^{(a)}\\|$. In this sense, we confirm that reciprocal orthogonality is approximately satisfied.
> > >
> > > Furthermore, in Figure 8 we compute the cosine similarity between $\\boldsymbol{W}\_\\ell^{(a)} \\boldsymbol{z}\_{\\ell-1}^{(c)}$ and  $\\boldsymbol{W}\_\\ell^{(a)} \\boldsymbol{z}\_{\\ell-1}^{(a)}$, and we observe that, as we increase the width, this cosine similarity approaches 1 across all layers. This behavior agrees with the theoretical prediction when LEWC and reciprocal orthogonality hold simultaneously.
> > >
> > > Thus, we verify LEWC and reciprocal orthogonality through multiple indicators, rather than inferring them solely from the exponential decay of activation norms.
> > >
> > > ## Distinguishing our explanation from a simpler explanation based on flatness
> > >
> > > As you noted, it is natural and important to ask whether some aspects of LMC could be explained simply by high flatness. We kept this point in mind and examined it from the perspective of flatness in Appendix D. Here we clarify our position.
> > >
> > > First, our main focus is on the landscape of the calibrated loss. In contrast, standard flatness measures (for example, equation (1) in Andriushchenko et al. (2023)) are defined for the uncalibrated loss, and quantify flatness via the variation of the loss under random Gaussian perturbations.
> > >
> > > In our experiments, we observe the following overall pattern:
> > >
> > > - In terms of the uncalibrated loss, even when the width is sufficiently large, the loss landscape is still not flat enough, and LMC does not hold.
> > > - In contrast, when we consider the calibrated loss, we observe LMC between the two models.
> > > - Our claim is that this flattening of the calibrated loss between the two models can be theoretically explained by assuming LEWC and reciprocal orthogonality.
> > >
> > > If LMC could be explained solely by an increase in flatness, then increasing the width should also naturally lead to LMC in terms of the uncalibrated loss. However, in practice, flatness in this sense is not sufficient, and LMC is not observed unless calibration is introduced. Therefore, we believe there is a gap in the explanation that "LMC occurs simply because increasing the width makes the landscape flat."
> > >
> > > At the very least, among the various experiments we conducted, we did not observe any strong contradictions with LEWC, and we believe that LEWC provides information that goes beyond what can be inferred from flatness alone.
> > >
> > > Maksym Andriushchenko, Francesco Croce, Maximilian Müller, Matthias Hein, and Nicolas Flammarion. 2023. A modern look at the relationship between sharpness and generalization. In Proceedings of the 40th International Conference on Machine Learning (ICML'23), Vol. 202. JMLR.org, Article 36, 840–902.

---

> > > > ### Author Response · Authors · 2025-11-28
> > > >
> > > > ## Behavior outside the interpolation interval (e.g., $\\lambda = 1.5, -0.5$)
> > > >
> > > > We also performed the experiment you suggested. Concretely, given two trained models $\\boldsymbol{\\theta}\_a$ and $\\boldsymbol{\\theta}\_b$, we evaluated the loss and accuracy at the point $\\boldsymbol{\\theta}\_a - (\\boldsymbol{\\theta}\_b - \\boldsymbol{\\theta}\_a) = 2\\boldsymbol{\\theta}\_a - \\boldsymbol{\\theta}\_b$, which lies outside the segment between $\\boldsymbol{\\theta}\_a$ and $\\boldsymbol{\\theta}\_b$. This corresponds to $\\lambda = 2$. For this experiment we used the widest models considered in the paper (MLP with $16 \\times$ width, VGG-11 with $16\\times$ width, and ResNet-20 with $32\\times$ width). The results are as follows:
> > > >
> > > > |                                                                        |   MLP MNIST |   MLP FMNIST |   VGG-11 CIFAR-10 |   ResNet-20 CIFAR-10 |
> > > > |:-----------------------------------------------------------------------|------------:|-------------:|------------------:|---------------------:|
> > > > | Loss w/o calibration ($\\boldsymbol{\\theta}\_a$)                         |    0.074809 |     0.346761 |          0.322835 |             0.231201 |
> > > > | Accuracy ($\\boldsymbol{\\theta}\_a$)                                     |   97.975    |    87.7208   |         89.4083   |            92.4792   |
> > > > | Loss w/ calibration ($\\boldsymbol{\\theta}\_a$)                          |    0.066467 |     0.347168 |          0.318685 |             0.226847 |
> > > > | Loss w/o calibration ($2\\boldsymbol{\\theta}\_a - \\boldsymbol{\\theta}\_b$) |    0.527832 |     3.72656  |          0.964518 |             0.7736   |
> > > > | Accuracy ($2\\boldsymbol{\\theta}\_a - \\boldsymbol{\\theta}\_b$)           |   97.8208   |    86.4458   |         88.7542   |            91.4417   |
> > > > | Loss w/ calibration ($2\\boldsymbol{\\theta}\_a - \\boldsymbol{\\theta}\_b$) |    0.07371  |     0.463135 |          0.356364 |             0.268392 |
> > > >
> > > > The first three rows correspond to $\\boldsymbol{\\theta}\_a$, and the last three rows correspond to $2\\boldsymbol{\\theta}\_a - \\boldsymbol{\\theta}\_b$.
> > > >
> > > > From these results, we observe that:
> > > >
> > > > - For the uncalibrated loss, the loss at $2\\boldsymbol{\\theta}\_a - \\boldsymbol{\\theta}\_b$ becomes much worse compared to $\\boldsymbol{\\theta}\_a$, so the landscape is clearly not flat in this sense.
> > > > - On the other hand, the degradation in accuracy and calibrated loss is relatively modest, and in particular the accuracy remains quite high.
> > > >
> > > > Here we would like to emphasize that LEWC is a theory that is intended to describe the behavior for $\\lambda \\in [0,1]$. Thus, the reason why the accuracy is preserved at points corresponding to $\\lambda = 2$ or $\\lambda = -1$, i.e., outside the interpolation interval, lies outside the direct scope of the theory developed in this paper. However, these results do not support a simple explanation purely in terms of flatness either. If increasing the width merely made the entire loss landscape flat, then even in terms of the uncalibrated loss, the loss at $2\\boldsymbol{\\theta}\_a - \\boldsymbol{\\theta}\_b$ should not deteriorate so drastically, which is contrary to what we observe.
> > > >
> > > > The fact that the calibrated loss appears relatively flat even outside the interval is something our current theory does not fully explain, and we see it as an interesting direction for future work. At the same time, this does not contradict the role of LEWC in explaining LMC between the two models within $\\lambda \\in [0,1]$. Rather, we interpret it as suggesting that multiple mechanisms may coexist in shaping the landscape of the calibrated loss.

---

### Author Response · Authors · 2025-11-26

We thank all reviewers for their thoughtful comments. In response, we revised the manuscript accordingly. All modifications are highlighted in red.

Here, we address a common question raised by multiple reviewers regarding the relationship between LLFC and LEWC.

A key difference between LLFC and LEWC is that LLFC is a property that arises when the weights of two models are sufficiently close, whereas LEWC is a property that arises when the weights of two models are different, namely approximately orthogonal. This distinction can be explained by focusing on the difference between commutativity and reciprocal orthogonality, which are the respective sufficient conditions that distinguish LLFC from LEWC.

Below we explain this point in more detail using equations. Let $\\boldsymbol{\\theta}_a$ and $\\boldsymbol{\\theta}_b$ be the two models, and for simplicity assume that biases can be ignored. Commutativity for $\\ell \\geq 1$ requires

$
\\boldsymbol{W}\_\\ell^{(a)} \\boldsymbol{z}\_{\\ell-1}^{(a)} + \\boldsymbol{W}\_\\ell^{(b)} \\boldsymbol{z}\_{\\ell-1}^{(b)} = \\boldsymbol{W}\_\\ell^{(a)} \\boldsymbol{z}\_{\\ell-1}^{(b)} + \\boldsymbol{W}\_\\ell^{(b)} \\boldsymbol{z}\_{\\ell-1}^{(a)}
\\Leftrightarrow
(\\boldsymbol{W}\_{\\ell}^{(a)} - \\boldsymbol{W}\_{\\ell}^{(b)} ) (\\boldsymbol{z}\_{\\ell-1}^{(a)} - \\boldsymbol{z}\_{\\ell-1}^{(b)}) = 0.
$

This condition clearly holds when all weights match, that is, when $\\|\\boldsymbol{W}\_\\ell^{(a)} - \\boldsymbol{W}\_\\ell^{(b)}\\| = 0$. Zhou et al. stated that this explains why WM can find permutations that make LMC hold, since the objective of WM is to search for permutations that minimize the distance between the weights of two models. In spawning, it is also known that LMC holds between solutions obtained by branching from a partially trained initialization and then continuing SGD independently. This is because the first training steps before branching keep the two models close, which promotes commutativity. Zhou et al. experimentally verified that commutativity holds in the spawning setting as well. In contrast, reciprocal orthogonality clearly does not hold when the two weight matrices match. Specifically, it requires $\\boldsymbol{W}\_\\ell^{(a)} \\boldsymbol{z}\_{\\ell-1}^{(b)} = 0$ and $\\boldsymbol{W}\_\\ell^{(b)} \\boldsymbol{z}\_{\\ell-1}^{(a)} = 0$, but if the weight matrices are equal and biases are negligible, then the activations are also equal, and these conditions are not satisfied. This shows that commutativity asks the two models to be sufficiently close, whereas reciprocal orthogonality asks them to be sufficiently different, namely approximately orthogonal, so the two requirements are incompatible. For example, in Figures 7 and 11(a) of the current manuscript we showed that without permutations, commutativity fails while reciprocal orthogonality holds, which reflects that the two models remain sufficiently distinct. Conversely, when models are aligned using WM discovered permutations, or in the spawning setting, the weight matrices are expected to be close. In such cases, commutativity is likely to hold, whereas reciprocal orthogonality is unlikely to be satisfied.

This incompatibility implies that the reason for LMC identified in our paper is fundamentally different from the LLFC based explanation of LMC in Zhou et al. Intuitively, under commutativity, permutations make the two models sufficiently close, so the merged model also remains close to each of the original models, and its output stays similar to theirs. In contrast, in our explanation, the two models have weight matrices that are highly different, namely approximately orthogonal, which allows the merged model to embed the two functions independently.

This distinction also explains why inverse temperature calibration is necessary under LEWC. LLFC requires the two weight matrices to be close, so the norm of the merged weight matrix is similar to that of the original models. In contrast, LEWC requires the two weight matrices to be orthogonal, so the norm of the merged weight matrix is strictly smaller than that of each original model. Since this reduction occurs at every layer, the output norm decays exponentially with depth. This provides the reason for the exponential decay of activation norms under LEWC.

We added the above discussion in Appendix C.

---

### Meta-Review · Area_Chair_7ANu · 2025-12-26

**Summary:**

This work investigates the phenomenon known as  linear mode connectivity (LMC). Prior studies suggest that achieving LMC requires both permutation search and sufficiently wide models, as increased width enlarges the space of viable permutations. This work  empirically demonstrates  that widening the models alone is sufficient to achieve LMC when combined with proper softmax temperature calibration. This effect is explained by the observation that  each layer of the merged model behaves as an exponentially weighted combination of the corresponding layers of the original models (referred to as layerwise exponentially weighted connectivity). This depends on two conditions, namely, weak additivity and reciprocal orthogonality.

**Reviewer Concerns:**

All reviewers found the novel observations interesting. Weaknesses they saw were  that the reported observations could be explained by functional redundancy and flatness induced by widening the networks, experiments were just done on smaller datasets and networks, the work has not been situated nicely in the existing literature on overparmetized networks, week theory-experiment alignment, and missing practical implications.  Authors added new experiments  (showing that flattness alone is not sufficient as explanation) and convincingly reacted to most weaknesses in the rebuttal. It is true that no practical implications are directly analysed but this is true for most works analysing LMC. Moreover, in my opinion barriers should be analysed in the loss landscape (in which optimization takes place) and not with respect to accuracy. But this is also something that is frequently done in the previous line of work. Therefore, the results are nevertheless interesting and I still recommend acceptance.

**Reviewer Scores:**

Reviewer RKhX indicated that they would rise his score if (4) the experiments they suggested were added. Their main daught was that the reported observations could be explained by functional redundancy and flatness induceed by widening the networks. In the rebuttal the authors reported results from newly added experiments and presented arguments that emphasised that the induced flatness is likey not the reason for their observation. Experiments on larger networkts and architectures with more modern aticvatio functions have not been conducted. Nevertheless, I assume, that the reviewer might have raised their score due to the experiments invesigating the effect of flatness.

Reviewer 3r8c indicated that they would consider rising their score (4) if the authors can situate the implications of their work better within the existing literature. The authors have extended the discussion on related work specifically focussing on overparametrized neural networks. I am not an expert for this field but the discussion sounds convincing so I estimate that the reviewer would have increased their score.

Reviewer 8AS9 did reply before the bug and stated that he will keep his score (6).

Reviewer EvXf only listed as a weakness that the experiments where run with quite high weightdecay and authors replied to his questions. So I expect, that the reviewer would have at least kept his score

---

### Decision · Program_Chairs · 2026-01-26

Accept (Poster)